# Warm mid-Pliocene conditions without high climate sensitivity: the CCSM4-Utrecht (CESM 1.0.5) contribution to the PlioMIP2

Michiel L. J. Baatsen[1], Anna S. von der Heydt[1,2], Michael A. Kliphuis[1], Arthur M. Oldeman[1], and Julia E. Weiffenbach[1]

[1]Institute for Marine and Atmospheric research Utrecht (IMAU), Department of Physics, Utrecht University, Utrecht, The Netherlands.
[2]Centre for Complex Systems Science, Utrecht University, Utrecht, The Netherlands

**Correspondence:** Michiel Baatsen (m.l.j.baatsen@uu.nl)

**Abstract.** We present the Utrecht contribution to the Pliocene Model Intercomparison Project Phase 2 (PlioMIP2), using the Community Earth System Model version 1.0.5 (*CCSM4-Utr*). Using a standard pre-industrial configuration and the *enhanced* PlioMIP2 set of boundary conditions, we perform a set of simulations at various levels of atmospheric $pCO_2$. This allows us to make an assessment of the mid-Pliocene reference ($Eoi^{400}$) climate versus available proxy records and a pre-industrial control ($E^{280}$), as well as to determine the sensitivity to different external forcing mechanisms.

We find that our simulated Pliocene climate is considerably warmer than the pre-industrial reference, even under the same levels of atmospheric $pCO_2$. Compared to the $E^{280}$ case, our simulated $Eoi^{400}$ climate is on average almost 5°C warmer at the surface. Our $Eoi^{400}$ case is among the warmest within the PlioMIP2 ensemble and only comparable to the results of models with a much higher climate sensitivity (i.e. CESM2, EC-Earth3.3, and HadGEM3). This is accompanied by a considerable polar amplification factor, increased globally averaged precipitation and greatly reduced sea ice cover with respect to the pre-industrial reference. In addition to radiative feedbacks (mainly surface albedo, $CO_2$, and water vapour) a major contribution to the enhanced Pliocene warmth in these simulations is the warm model initialisation followed by a long spin-up, as opposed to starting from pre-industrial or present-day conditions. Added warmth in the deep ocean is partly the result of using an altered vertical mixing parametrisation in the Pliocene simulations, but this has a negligible effect at the surface. We find a stronger and deeper Atlantic Meridional Overturning Circulation (AMOC) in the $Eoi^{400}$ case, but the associated meridional heat transport is mostly unaffected. In addition to the mean state, we find significant shifts in the behaviour of the dominant modes of variability at annual to decadal timescales. The $Eoi^{400}$ ENSO amplitude is greatly reduced (-68%) versus the $E^{280}$ one, while the AMOC becomes more variable. There is also a strong coupling between AMOC strength and North Atlantic SST variability in the $Eoi^{400}$, while North Pacific SST anomalies seem to have a reduced global influence with respect to the $E^{280}$ through the weakened ENSO.

## 1 Introduction

The Pliocene Model Intercomparison Project Phase 2 (PlioMIP2; Haywood et al. 2016) aims at simulating the mid-Piacenzian Warm Period (mPWP; 3.264 to 3.025 Ma; Haywood et al. 2013a) of the mid-Pliocene using a suite of global numerical climate models. During this interval, the Earth's climate saw warm and relatively stable conditions with atmospheric $CO_2$ levels similar to those seen today. Being relatively recent in the planet's geologic history, geographical boundary conditions during the mid-Piancenzian were similar to the present. Therefore, the simulated climatic conditions for this specific time interval may serve as a suitable analogue of what to expect in the next century (Burke et al., 2018).

Previous modelling efforts within PlioMIP1 resulted in globally averaged near surface air temperatures between 1.8 and 3.6 °C warmer than those seen today (Haywood et al., 2013b), for the mid-Pliocene. The pattern of warming showed clear polar amplification, resulting in a reduced equator-to-pole temperature gradient. However, this reduction was still considerably smaller than suggested by various temperature proxy records and models mostly failed to reproduce the strong warming pattern over the North Atlantic Ocean. The simulated mid-Pliocene warming response was found to be mostly a result of increased atmospheric $CO_2$, with surface albedo feedbacks the primary driver behind polar amplification.

In addition to an extended model ensemble, PlioMIP2 offers the use of model boundary conditions from the Pliocene Research, Interpretation and Synoptic Mapping version 4 (PRISM4; Dowsett et al. (2016)). This includes updated estimates of vegetation cover, coastlines, and topography as well as reduced ice sheet cover compared to PRISM3 (Dowsett et al., 2010). As a result, models now show an average warming response of 1.7 to 5.2 °C to these mid-Pliocene boundary conditions. In addition, most models have a mid-Pliocene temperature pattern that better resembles that of proxy reconstructions, particularly with warmer high latitude and North Atlantic regions compared to PlioMIP1 (Haywood et al., 2020).

Here, we present the results of a set of simulations using the Community Earth System Model (CESM) version 1.0.5 within the PlioMIP2 effort. A detailed description of our model configuration, as used in these experiments, is provided in Section 2. This is followed by a general discussion of the PlioMIP2 experimental design and naming conventions in Section 3, together with an overview of our specific set of model experiments. The results of these are presented in Section 4, in which we assess climate sensitivity (4.1), the general mPWP conditions in our simulations (4.2), meridional overturning and heat transports (4.3), a comparison to proxy records (4.4), sensitivity to the applied PlioMIP2 model boundary conditions (4.5), and modes of internal variability in the simulated climate (4.6).

## 2 Model description

### 2.1 The CESM 1.0.5

The Community Earth System Model (Hurrell et al., 2013) is a fully coupled atmosphere-land-ice-ocean general circulation model (GCM) that was developed at the National Center for Atmospheric Research (NCAR) in Boulder, Colorado. For use in palaeoclimate modelling, version 1.0.5 of the CESM is a suitable choice motivated by a trade-off between increasing model complexity and computational cost. This version of the CESM is equivalent to the last version of its predecessor, the Com-

munity Climate System Model (CCSM4; Blackmon et al. 2001; Gent et al. 2011). In other PlioMIP2 studies, our model simulations are therefore referred to as *CCSM4-Utr*.

## 2.2 Atmosphere

The atmospheric component of the CESM is the Community Atmosphere Model (CAM4; Neale et al. 2013) which uses a finite volume dynamical core. The model grid has a nominal horizontal resolution of $2°$ ($2.5° \times 1.9°$; $144 \times 96$ grid cells). 26

vertical levels extend upward to 2 hPa, using a hybrid sigma vertical coordinate. In this configuration, the model has a warming response of $3.17°C$ per doubling of $CO_2$ starting from pre-industrial conditions (Baatsen et al., 2020), which is very similar to the reported value of $3.14°C$ by Bitz et al. (2012), and higher than the $\sim 2.5°C$ in CCSM3; Kiehl et al. 2006).

In accordance to the PlioMIP2 protocol (Haywood et al., 2016), atmospheric concentrations other than that of $CO_2$ are kept at their pre-industrial levels: 671ppb $CH_4$, 270ppb $N_2O$, and no CFCs. The solar constant is kept at 1361.27 $Wm^{-2}$ in all of

our model experiments. Astronomical orbital parameters are set to their present-day configurations for pre-industrial as well as Pliocene simulations, again as suggested by the PlioMIP2 protocol. Atmospheric aerosols are fixed using a pre-industrial climatology for the pre-industrial cases. An adjusted climatology is used for the Pliocene cases, resulting from a Bulk Aerosol Model simulation to incorporate the effect of the PRISM4 boundary conditions.

## 2.3 Land

The physical, chemical and biological processes taking place on land are represented in the Community Land Model (CLM4; Oleson et al. 2010; Lawrence et al. 2011). A static rather than dynamic vegetation model is used here to avoid runaway feedback effects, which can become an issue especially in warm greenhouse climates (e.g. dieback of vegetation at low latitudes; Loptson et al. 2014; Herold et al. 2014). Either the pre-industrial biomes or the PRISM4 megabiomes (see also Supplementary Table S1) are translated into fractions of the corresponding CLM4 plant functional types (PFTs), from which a set of monthly forcing

files is ultimately used in the model. Fresh water runoff is treated by a simple river routing scheme, in which all runoff is transported to one of the surrounding 8 model grid cells until the ocean is reached. The direction is determined by the local topography gradient and manually adjusted where the runoff scheme would otherwise form closed loops. Within the CLM4, land ice is implemented as bare soil with a given surface elevation and its specific surface properties (e.g. albedo, evaporation and run-off). As suggested by Haywood et al. (2016), the static land ice configuration used here is based on the results of

previous modelling efforts in PlioMIP1. This includes a greatly reduced Greenland Ice Sheet compared to the present day, as well as an absent West-Antarctic Ice Sheet.

## 2.4 Ocean

The CESM1 uses the Los Alamos National Laboratory (LANL) Parallel Ocean Program version 2 (POP2; Smith et al. 2010) for the ocean model component. The standard configuration is applied here, with a nominal $1°$ ($1.25° \times 0.9°$) horizontal resolu-

tion on a curvilinear grid placing the northern pole over Greenland. The POP2 model is set up with 60 vertical layers of varying

thickness between 10m near the surface and 250m at greater depth. Horizontal viscosity is considered anisotropic (Smith and McWilliams, 2003) and horizontal tracer diffusion follows the parameterisation of Gent and McWilliams (1990). The model further uses the KPP-scheme to determine vertical mixing coefficients (Large et al., 1994). More information and discussion on the ocean model physics and parameterisations can be found in Danabasoglu et al. (2008, 2012).

The sea ice component consists of the LANL Community Ice Code version 4 (CICE4; Hunke and Lipscomb 2008). For simplicity, sea ice only forms when the sea surface cools down to -1.8°C, after which its dynamical behaviour (e.g. melt and advection) is treated by the model specifically.

In all of our model experiments considered here, tidal mixing and overflow parameterisations are switched off. The pre-industrial reference uses a uniform background vertical diffusivity, set at 0.16 cm$^2$s$^{-1}$. In contrast, the Pliocene simulations

have a vertically varying background vertical diffusivity determined by: $\kappa_w = vdc1 + vdc2 \tan^{-1}((|z| - dpth)linv)$, where: $vdc1 = 0.524$ cm$^2$s$^{-1}$, $vdc2 = 0.313$ cm$^2$s$^{-1}$, $dpth = 1000$m (the reference depth) and $linv = 4.5 \cdot 10^{-3}$ m$^{-1}$ (the inverse scaling length). $\kappa_w$ thus varies between 0.1 cm$^2$s$^{-1}$ at the surface and 1 cm$^2$s$^{-1}$ at the bottom of the ocean. The latter is done to be consistent with other palaeo-climate simulations using the same model set-up, e.g. Baatsen et al. (2020). To study the effect of the different mixing parameter settings, we carry out an additional pre-industrial simulation using the same oceanic configuration

as for the Pliocene cases (see Section 3, Figures S4, S7, and S9 in the supplementary material). The enhanced vertical mixing in the deep ocean results in an overall warming of the deep ocean at the expense of a slight cooling at upper levels. Near the surface, globally averaged temperatures are near identical but some regional changes are seen as a result of the different vertical mixing schemes. For completeness, another pre-industrial simulation is carried out using the standard configuration in which the tidal mixing and overflow parameterisations are switched on. The results show no significant temperature differences seen

at any depth level and thus suggests that the combined effect of these parameterisations is negligible within the scope of this study and the specific model configuration used here.

## 3 PlioMIP2 experiments

### 3.1 PRISM4 boundary conditions

Following the methodology outlined by Haywood et al. (2016), we perform a set of pre-industrial and Pliocene simulations

using the CESM1.0.5. As we carry out a number of different model experiments, we will refer to these as either *pre-industrial* or *Pliocene* (rather than mid-Piacenzian) regardless of atmospheric pCO$_2$. For Pliocene cases we incorporate the set of enhanced boundary conditions based on the PRISM4 (Dowsett et al., 2016). These include altered topography and bathymetry, coastlines, land surface properties (i.e. vegetation, soil type and ice sheet coverage) and atmospheric composition with respect to pre-industrial conditions. Our Pliocene model geography is implemented by applying the PRISM4 topography and bathymetry

anomalies to the pre-industrial reference, after re-gridding onto the specific model resolution used here. The PRISM4 land-sea mask is then also interpolated onto the model grid and applied to the Pliocene model geography. A manual check of the entire model grid is done, altering land and ocean cells where needed and making sure that any marine passages are either at least 2 cells wide when open, or closed if needed. The resulting model geography for the Pliocene simulations is shown in Figure 1

and compared to that of the pre-industrial reference. Corresponding vegetation cover and aerosol optical depth can be found in Figure S1 of the supplementary material.

Some of the main aspects of the Pliocene model geography with respect to the pre-industrial reference include a closure of

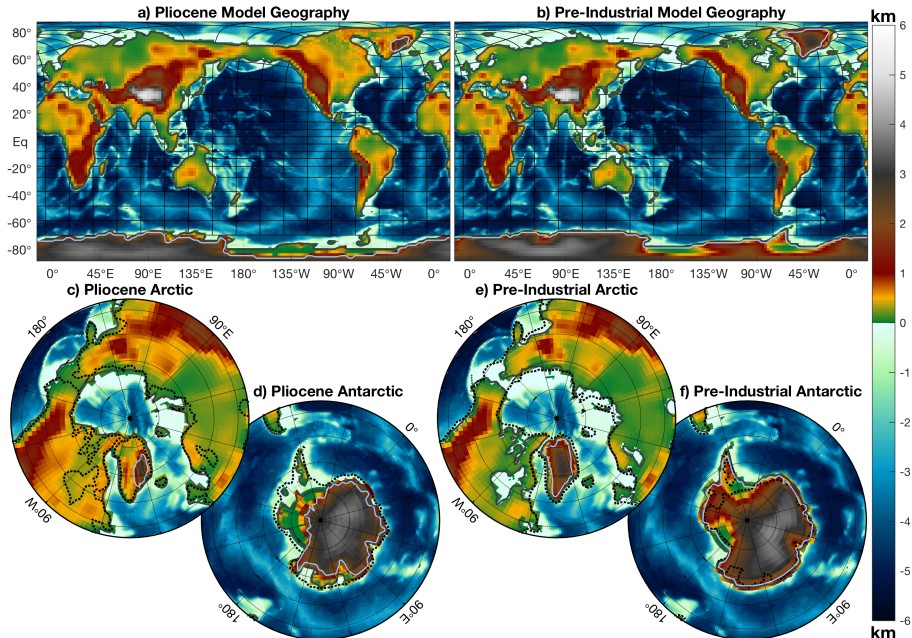

**Figure 1.** Model geography as applied in our different **a)** Pliocene and **b)** pre-industrial cases. Both the bathymetry and topography are shown at the native model grid, with the CAM4 rectangular grid superimposed on the finer POP2 curvilinear grid wherever the land fraction is at least 0.5 (also indicated by the thick grey contour line). The light blue contour line shows the extent of land ice, as imposed by the land use distribution. **c,e** Northern and **d,f** Southern polar stereographic views Pliocene and pre-industrial model geography.

both the Bering Straight and Northwest Passage, making the Arctic Ocean much more isolated from the other ocean basins. Prior to large-scale glaciation on the Northern Hemisphere during the Quaternary period, a considerable part of the northern Eurasian continental shelf as well as the Hudson Bay area were exposed. Furthermore, large parts of the Maritime Continent are also considered to have been above sea level during the mid-Pliocene. Finally, there was significantly less land ice cover compared to the present, covering only part of Southeast Greenland and East Antarctica. Note that the Gibraltar straight is open while the Panama Gateway is closed in all of our simulations (this is obscured in the figures by the superposition of the atmospheric model grid onto the bathymetry).

## 3.2 Experimental design and model spin-up

In accordance with the PlioMIP2 guidelines we performed both a pre-industrial reference and Pliocene control *core* simulation, which are referred to as $E^{280}$ and $Eoi^{400}$, respectively (following the naming conventions of Haywood et al. 2016). In addition to the pre-industrial reference we added two *climate sensitivity* simulations, with a doubling ($E^{560}$) and quadrupling ($E^{1120}$) of

atmospheric $CO_2$. Two more sensitivity experiments were added to the Pliocene control, in which we applied pre-industrial (Eoi$^{280}$) and doubled (Eoi$^{560}$) $CO_2$ levels. This enables us to make a complete assessment of the model's sensitivity to either Pliocene boundary conditions or radiative forcing, for the full range of possible reference states. Besides altered model boundary conditions, the Pliocene simulations differ from the pre-industrial ones as a result of the oceanic vertical mixing parameters. Another *mixing sensitivity* experiment is thus added as a continuation from the E$^{280}$ simulation, using the same ocean mixing configuration for the ocean as in the Pliocene cases, referred to as E$^{280,P}$. Results of an additional (500-year) pre-industrial simulation with tidal mixing and overflow parameterisations switched on (E$^{280,S}$), are not used besides a technical robustness check and therefore not considered. See also Table 1 for an overview of the different simulations.

The pre-industrial reference simulation is initialised using present-day (year 2000) temperature and salinity fields from the

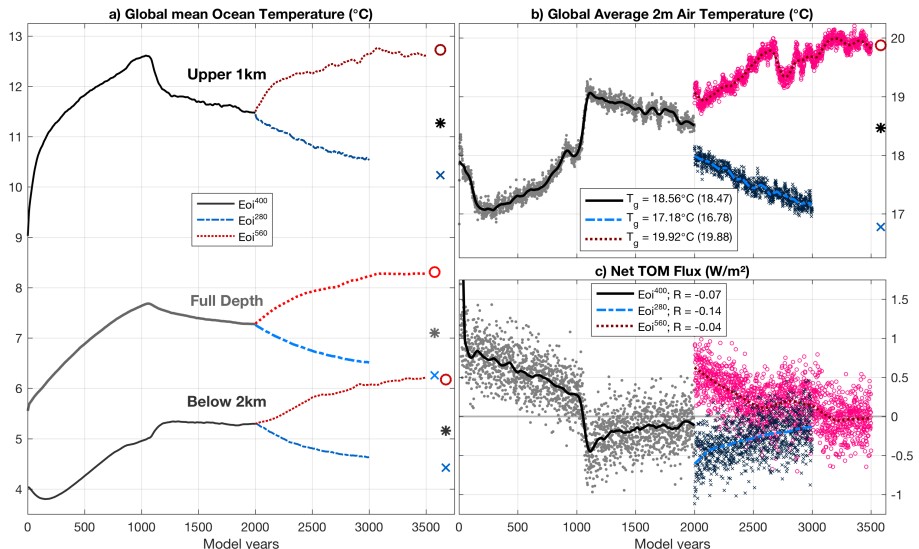

**Figure 2.** Time series of globally averaged temperatures for the entire length of our 3 different Pliocene simulations: Eoi$^{400}$ (black), Eoi$^{280}$ (blue), and Eoi$^{560}$ (red). Shown are the **a)** upper (dark), deep (medium), and full depth (light) ocean temperature, **b)** near surface air temperature, and **c)** globally averaged top of model (TOM) net radiative flux. Thick lines in b,c show the corresponding time series after applying a 100-year smoothing mask. The estimated equilibrium temperatures are indicated at the end in a,b, using large markers and the same colour convention. The globally averaged, mean temperature ($T_g$ in b) and net radiative flux (R in c) over the last 100 years are added in the legends (bracketed values for the estimated equilibrium).

PHC2 dataset (Steele et al., 2001). As these are slightly warmer than pre-industrial conditions, a long spin-up of $>3000$ years is carried out to equilibrate the full ocean. The remaining drift of volume-weighted average ocean temperature at the end of this simulation is brought down to $\sim 10^{-4}$ K/year (Table 1). In contrast, the Pliocene control simulation is started with a highly idealised ocean temperature and salinity distribution. The initial temperature is horizontally homogeneous while salinity is set to 35psu globally. The vertical profile of temperature decreases downward between 15°C at the surface and 4°C in the deep ocean. We thus apply similar initial conditions to those used in the Eocene simulations of Baatsen et al. (2020), but with cooler

deep ocean temperatures. The latter is done to match the volume-integrated, global ocean temperatures of Rosenbloom et al. (2013), who used the CCSM4 to model the mid-Pliocene within the PlioMIP1. This way, we start the model with a total ocean

heat content similar to previously found Pliocene conditions using a very similar model configuration. Still, we perform 2000 model years of spin-up to acquire a well-equilibrated oceanic state for our Pliocene control. An overview of the temperature evolution during this spin up along with the top of model radiative imbalance can be found in Figure 2, which shows substantial temperature adjustments. After an initial warming phase, an abrupt change in temperature trends is seen at $\sim$1000 years into the $\text{Eoi}^{400}$ simulation. This is the result of a shift in the oceanic circulation, associated with the formation of deep overturning

cells in the North Atlantic and Southern Ocean. These act to mix heat from the upper into the deep ocean, but also increase high latitude temperatures through enhanced meridional heat transport (see also Figure 7). Similar temperature trends were seen by Chandan and Richard Peltier (2017), especially considering their *POP1 type* vertical mixing simulations, in which they implement an identical ocean set-up to our mid-Pliocene cases (and $\text{E}^{280,\text{P}}$) yet with a cooler initial state. Additional information on the spin-up behaviour of the set of model simulations presented here can be found in Figures S2 and S3 of the supplementary

material.

All three pre-industrial sensitivity experiments are started from the equilibrated state at the end of the $\text{E}^{280}$ control simulation. The $\text{E}^{560}$ and $\text{E}^{1120}$ cases are continued for another 1000 and 2000 model years, respectively, while the $\text{E}^{280,\text{P}}$ is continued for 2300 model years (see also Figure S4 of the supplementary material). The $\text{E}^{560}$ and $\text{E}^{1120}$ simulations still have substantial drifts in deep ocean temperature, but much less near the surface. Using the transient behaviour, however, we can estimate the

actual equilibrium temperatures (see section 3.3) and therefore the climate sensitivity of the model within this specific set-up. In the global average, the effect of altered ocean mixing parameters is quite clear: there is no temperature change at the surface, but the upper ocean cools slightly while the deep ocean warms. The more efficient downward mixing of heat leads to an average warming of the total water column (by $\sim$0.8$^\circ$C) in the extrapolated equilibrium. The $\text{Eoi}^{280}$ and $\text{Eoi}^{560}$ simulations are initialised from the $\text{Eoi}^{400}$ control at year 2000 and continued for 1000 and 1500 years, respectively. As their initial radiative

forcing is relatively small compared to a full doubling of $CO_2$, it does not take as long for the temperature drifts to become small. The $\text{Eoi}^{560}$ simulation is longer compared to the $\text{Eoi}^{280}$ one as it was showing substantial internal variability after about 700 years owing to large (i.e. $>$10Sv) fluctuations in the Atlantic Meridional Overturning Circulation strength (Figure S5 in the supplementary material).

### 3.3   Data analysis and methods

Within the PlioMIP2 database, model fields from the last 100 model years of the $\text{E}^{280}$, $\text{E}^{560}$, $\text{Eoi}^{280}$, $\text{Eoi}^{400}$, and $\text{Eoi}^{560}$ case are publicly available. We use the same 100 years here for our analyses and to calculate time means in order to best match the results of other studies. Some more extensive data time series are considered as well, but only when noted specifically. The overview of modelled atmospheric and oceanic conditions given in Section 4.2 is based on climatologies using the same 100 years at the end of the $\text{E}^{280}$ and $\text{Eoi}^{400}$ runs. From the $\text{E}^{280,\text{P}}$ simulation we use 200-year averages instead (i.e. 4400–4600)

as it exhibits some long-term fluctuations (see Figure S4 in the supplementary material). In addition to the model data presented here, we make use of satellite-based passive microwave sea ice concentration data (Peng et al., 2013). We consider the

1988–2000 mean sea ice concentration, excluding more recent years in which a rapid decline of Arctic sea ice was observed. We also use mid-Pliocene (more specifically the KM5c interglacial period) SST proxies from McClymont et al. (2020). This record includes absolute SST estimates using the newly calibrated $U_{37}^{K'}$ and Mg/Ca proxies, as well as SST anomalies which
are calculated with respect to the 1870–1899 ERSST reanalysis (Huang et al., 2017).

In addition to direct climatological or annual mean fields at the end of each simulation, we also present a number of esti-mated equilibrium values using the complete time series of the according temperatures and radiative fluxes. Based on the method of Gregory et al. (2004), we linearly extrapolate the trend of a specific temperature measure towards a net zero top
of model (TOM) radiative flux. It should be noted that the original technique of Gregory et al. (2004) is designed to estimate equilibrium temperatures from relatively short (∼50 years) simulations following an initial perturbation. Using a much longer time series, we do include the effects of slower (e.g. ocean thermal adjustment; Baatsen et al. 2020; Farnsworth et al. 2019) and nonlinear (e.g. clouds; Bloch-Johnson et al. 2015; Knutti and Rugenstein 2015) feedbacks and therefore can acquire a better estimate of the actual equilibrium temperature (but also require a much longer simulation time). The net TOM radiation is cal-
culated by subtracting the (surface-weighted) globally averaged net outgoing longwave flux from the net incoming shortwave flux. A visualisation of the standard procedure for globally averaged near surface air temperature in the Eoi[280], Eoi[400], and Eoi[560] cases can be found in Figure S2 of the supplementary material. An extension towards upper (<1km), deep (>2km) and full-depth ocean temperatures from all of our simulations is also provided in Figure S3. As shown by Gregory et al. (2004), the equilibration of the globally averaged near surface air temperature from an initial shock in radiative forcing becomes linear
as a function of the net TOM radiative flux. However, their simulations use a slab ocean rather than a full-depth ocean model component. Using our model configuration, the equilibration is slightly non-linear instead, especially at the start of the simula-tion (see also Baatsen et al. 2020). Therefore, we exclude the first 100 or 250 years of data from this analysis for atmospheric or oceanic temperatures, respectively.

To get a quantitative assessment of the temperature contributions from different components in the radiative balance between our *sensitivity* simulations, we repeat the analysis of Hill et al. (2014) using an energy balance framework similar to Lunt et al. (2021); Heinemann et al. (2009). We adopt the radiative balance equation as:

$$S(1 - \alpha_p) + H + F_{cloud} + F_{CO_2} = \epsilon \sigma \tau^4,$$

where $S$ is the incoming solar radiation, $\alpha_p$ the planetary albedo, $H$ the meridional heat transport, $F_{cloud}$ the cloud radiative
forcing, $F_{CO_2}$ the radiative forcing resulting from a doubling of atmospheric pCO$_2$, $\epsilon$ the emissivity, $\sigma$ the Stefan-Boltzmann constant, and $\tau$ the surface temperature. This equation can be rearranged to determine the surface temperature, using the annual mean model fields from each simulation. The temperature effect of a single component between 2 different simulations can then be estimated by the temperature difference by adjusting this component in the energy balance equation. Note that in the reference temperature, both $F_{cloud}$ and $F_{CO_2}$ are 0. In contrast to Lunt et al. (2021), we consider the full 2D fields and then take
the zonal average of the resulting temperature difference. We also include the effects of surface albedo, shortwave/longwave

cloud forcing and $CO_2$ as components in the energy balance.

In our analysis of the ocean circulation, we consider the barotropic stream function (BSF) and meridional overturning stream function (MSF). Both of these are calculated from the monthly averaged horizontal flow fields and integrated as a function of latitude and either depth (BSF) or longitude (MSF), starting from the southern pole located on Antarctica. The last 100 years of each simulation are again used for the calculation of the BSF, but the last 500 years are used for the MSF (to avoid a possible influence of centennial-scale variability). From the MSF, we also determine the strength of the Atlantic meridional overturning circulation (AMOC). This is defined as the maximum of the overturning stream function below 1000m and north of 30°S, using only the flow field in the Atlantic Ocean.

We look at the temporal variability of SSTs through El Niño Southern Oscillation (ENSO), Atlantic/Pacific (Multi-)Decadal Variability (AMV/PMV) and the AMOC time series. ENSO is characterised by the Niño indices, taking the monthly average SST anomaly over 0–10 °S, 90–80°W (Niño 1+2) and 5°N–5°S, 170–120°W (Niño 3.4). To capture ENSO variability on decadal time scales as well, we use 200 rather than 100 years of monthly SST data. We subtract the 12-monthly climatological mean and apply a linear de-trending to each model grid point. The resulting Niño time series are then treated with a 5-month running mean.

The calculation of AMV/PMV indices requires the use of empirical orthogonal functions (EOFs), using 500 years of annual mean SST data as these processes occur on longer timescales (see e.g. Deser et al. 2012; Trenberth and Shea 2006, and https://climatedataguide.ucar.edu/data-type/climate-indices; date of last access: 17/08/2021). We apply a local de-trending to annual SST time series at each grid point by subtracting a 200-year smoothing spline. We then determine the first three spatial EOFs of the resulting SST anomalies for the North Atlantic (10–70 °N) and North Pacific (20–60 °N) Ocean. The AMV mode is then determined as the EOF of which the time series correlates best with the North Atlantic average SST (rather than the first EOF, as the variability may be dominated by e.g. ENSO or PMV). The PMV is simply the dominant EOF in the North Pacific Ocean, obtained from the annual SST anomalies. Patterns of SST variability corresponding to these modes can then be obtained by correlating the local SST anomalies to each of the specific mode time series.

We perform a spectral analysis of the different modes of variability, using a multi-tapered method (MTM) as it is more suitable for climatic time series of limited length (Ghil et al., 2002). This method is based on a standard Fourier analysis, but uses tapers to limit cut-off effects at the edges of the respective time series. We apply this technique, using 3 tapers and a bandwidth parameter of 2. All of the resulting power spectra are tested against a red noise null hypothesis, using 10.000 random AR1 surrogate time series. The median of these red noise power spectra is used to normalise those of the corresponding modes of variability, making the latter easier to interpret as well as consider their statistical significance using 90%, 95%, and 99% confidence levels. Finally, we also correlate the time series of different modes: AMV, PMV, AMOC and ENSO (now using 500 years of annual SST anomalies for the latter as well, rather than 200 years of monthly SST anomalies). These correlations are considered significant when their corresponding p-value is less than 0.05 (i.e. white noise null hypothesis).

**Table 1.** Overview of globally averaged observables and equilibration measures for all of our simulated cases, including simulation length (+ denotes a continuation from the above control case), mean annual 2m air temperature (MAT), average tropical temperature $MAT_T$, average polar temperature $MAT_P$, mean annual precipitation (MAP), top of model net radiative flux ($R_{TOM}$), sea surface temperature (SST), full depth ocean temperature (OT), and ocean temperature drift. Subscript $e$ denotes estimated equilibration values.

| Measure Simulation | Length (years) | MAT (°C) | $MAT_e$ (°C) | $MAT_T$ (°C) | $MAT_P$ (°C) | MAP (m yr$^{-1}$) | $R_{TOM}$ (W m$^{-2}$) | SST (°C) | OT (°C) | $OT_e$ (°C) | OT Drift (10$^{-4}$ K yr$^{-1}$) |
|---|---|---|---|---|---|---|---|---|---|---|---|
| **E$^{280}$** | **3100** | **13.85** | *13.82* | **25.32** | **−22.51** | **1.046** | **−0.002** | **18.42** | **3.04** | *3.04* | **−1.03** |
| E$^{280,P}$ | +2200 | 13.87 | *13.83* | 25.28 | −22.30 | 1.044 | +0.016 | 18.40 | 3.61 | *3.65* | +1.56 |
| E$^{560}$ | +1000 | 16.54 | *16.75* | 27.25 | −16.94 | 1.092 | +0.187 | 20.23 | 4.14 | *5.14* | +6.18 |
| E$^{1120}$ | +2000 | 19.86 | *20.16* | 29.52 | −9.52 | 1.156 | +0.214 | 22.66 | 6.66 | *8.15* | +6.89 |
| **Eoi$^{400}$** | **2000** | **18.54** | *18.47* | **28.02** | **−10.09** | **1.181** | **−0.073** | **21.89** | **7.28** | *7.09* | **−1.64** |
| Eoi$^{280}$ | +1000 | 17.17 | *16.78* | 27.01 | −12.98 | 1.154 | −0.139 | 20.86 | 6.53 | *6.25* | −3.19 |
| Eoi$^{560}$ | +1500 | 19.91 | *19.88* | 29.01 | −7.21 | 1.208 | −0.042 | 22.97 | 8.28 | *8.27* | +0.76 |

## 4 Results

### 4.1 Global averages and climate sensitivity

An overview of globally averaged temperatures, drifts and radiative balance at the end of each of our model simulations is provided in Table 1. These values, including some additional metrics of the vertical thermal distribution of the ocean, are visualised in Figure 3. Throughout the different cases, we see a consistent globally averaged equilibrium warming of ∼3°C per $CO_2$ doubling in the near surface atmosphere. This warming response reduces slightly to ∼2.5°C in the ocean. The globally averaged sea surface temperature (SST) increases by ∼2.1°C per $CO_2$ doubling. In addition to different surface feedbacks over land versus sea, the inhomogeneous distribution of land/sea (i.e. more land at high latitudes) acts to further differentiate between the average temperature over the land and sea surface. There is some nonlinear response towards higher $CO_2$ concentrations, consistent with e.g. Lunt et al. (2021); Baatsen et al. (2020); Caballero and Huber (2013), which is closely related to the corresponding radiative forcing in the model. Starting from the pre-industrial reference, a single $CO_2$ doubling results in an initial perturbation of 3.49 W m$^{-2}$ while a quadrupling of $CO_2$ induces 7.93 W m$^{-2}$, meaning that the radiative forcing of 4×$CO_2$ is 2.27 (rather than 2) times that of 2×$CO_2$ (Baatsen et al., 2020; Etminan et al., 2016). Using the corresponding deviation of the globally averaged near surface air temperature (2.93°C and 6.34°C in the E$^{560}$ and E$^{1120}$, respectively), we find a climate sensitivity parameter of 0.80–0.84 K W$^{-1}$ m$^2$. Using the E$^{1120}$ extrapolated temperature, we report an estimated equilibrium climate sensitivity (ECS) of 3.17°C per $CO_2$ doubling, noting the nonlinearity towards higher pCO$_2$. Comparing the Eoi$^{560}$- Eoi$^{280}$ temperatures, we find a difference of 3.2°C, indicating a similar ECS between the modelled pre-industrial and Pliocene states. The observation that the Pliocene ECS tends towards the higher regime of pre-industrial (i.e. E$^{1120}$- E$^{560}$

rather than $E^{560}$- $E^{280}$) suggests that the nonlinearity in ECS is a function of the reference temperature rather than $pCO_2$, as the $Eoi^{280}$ temperatures are comparable to the $E^{560}$ ones.

Strikingly, the applied mid-Pliocene model boundary conditions result in an average warming similar to a doubling of atmospheric $pCO_2$ regardless of background $CO_2$ level. Looking at the upper ($<1000$m) versus deep ($>2000$m) ocean average temperatures in Figure 3, the offset between pre-industrial and mid-Pliocene conditions seems depth-dependent. While the upper ocean is about $2°$C warmer in our mid-Pliocene simulations, the deep ocean is over $3.5°$C warmer compared to the pre-industrial cases. Much of this warming pattern can be explained by taking into account the extrapolated equilibrium temperatures of the $E^{280,P}$ simulation as well (crosses in Figure 3, see also Figure S4). The vertical redistribution of heat can be mostly attributed to the vertical mixing parameters in the mid-Pliocene model set-up. After correcting the pre-industrial temperatures with the $E^{280,P}$- $E^{280,}$ difference, the average offsets between pre-industrial and mid-Pliocene simulations again closely resembles that of a $CO_2$ doubling (i.e. $\sim2.5°$C; bracketed values in Figure 3). A more detailed look into the distribution of different temperature responses to only a change in atmospheric $pCO_2$ or other model boundary conditions is presented in Section 4.5.

Compared to the $E^{280}$ pre-industrial reference, the simulated climate of our $Eoi^{400}$ mid-Pliocene control is on average almost $5°$C warmer at the surface. This is accompanied by a $13\%$ increase in mean annual precipitation and a considerable polar amplification factor. The latter is found by comparing the change in average polar ($>66.5°$N/S) to tropical ($<23.5°$N/S) temperature: $MAT_P/MAT_T$ between different cases. This gives a polar amplification factor of 3.11 and 3.17 for the $E^{560}$ and $E^{1120}$, respectively, compared to the $E^{280}$. A slightly smaller value of 2.85 is found between the $Eoi^{560}$ and $Eoi^{280}$, but a much larger polar amplification factor of 4.75 is seen between the $Eoi^{400}$ and $E^{280}$, including the effect of changing the model boundary conditions. This value increases further to 5.78 ($Eoi^{280}$- $E^{280}$) or 5.42 ($Eoi^{560}$- $E^{560}$) between Pliocene and pre-industrial cases at equal atmospheric $pCO_2$, indicating a strongly enhanced temperature response towards high latitudes to the applied Pliocene model boundary conditions.

As shown in Figure 3 and Table 1, the full depth average of the oceans is about $2.5°$C warmer in our Pliocene simulations compared to the equivalent pre-industrial case at equilibrium. Differences in land ice cover and vegetation type, but also snow and sea ice coverage provide a substantial contribution to the modelled Pliocene warmth (see also Table S2 in the supplementary material). Shortwave surface fluxes alone account for a $\sim6$ W m$^{-2}$ net forcing in the globally averaged radiative balance, comparing Pliocene to pre-industrial cases at equal $pCO_2$ ($Eoi^{280}$- $E^{280}$: 6.6 W m$^{-2}$; $Eoi^{560}$- $E^{560}$: 5.4 W m$^{-2}$). The responsible surface albedo feedback thus plays a primary role, decreasing slightly towards a warmer reference state. Although small, shortwave cloud feedbacks ($Eoi^{280}$- $E^{280}$: -1.2 W m$^{-2}$; $Eoi^{560}$- $E^{560}$: 0 W m$^{-2}$) counteract the reduced effect of surface albedo towards warmer states, helping to explain why we see a similar climate sensitivity across all of our model cases. Longwave radiative fluxes play a similar role to enhance Pliocene warmth in our simulations, mainly through the lapse rate and water vapour feedback. A warmer ocean surface and atmosphere lead to an increase in the total column water vapour, which acts as a greenhouse gas. In contrast to shortwave fluxes, this effect becomes larger towards warmer conditions ($Eoi^{280}$- $E^{280}$: 11.4 W m$^{-2}$; $Eoi^{560}$- $E^{560}$: 12.1 W m$^{-2}$), but is again mitigated by small longwave cloud feedbacks ($Eoi^{280}$- $E^{280}$: 0.7 W m$^{-2}$; $Eoi^{560}$- $E^{560}$: 0.2 W m$^{-2}$).

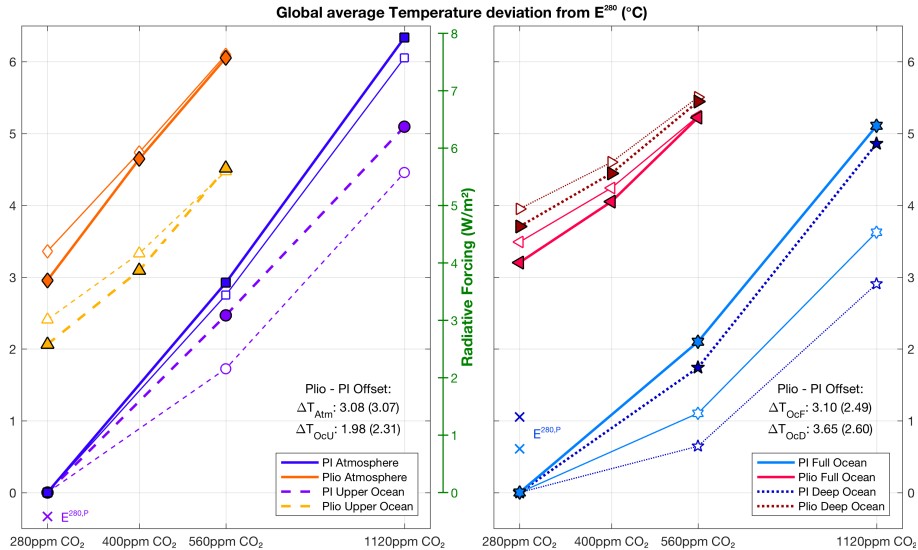

**Figure 3.** Overview of global average temperature in all of our different pre-industrial and Pliocene cases, compared to the $E^{280}$. Open markers, connected by thin lines, show the time mean at the of each simulation. Filled markers, connected by thick lines, show the corresponding equilibrium values. Cool colours are used for pre-industrial (*PI*) cases, warm colours for the Pliocene (*Plio*) ones. We consider 4 different temperature measures: near surface air temperature, upper ocean, full ocean, and deep ocean surface/volume weighted averages. Crosses indicate extrapolated temperatures from the $E^{280,P}$ case. Pliocene–pre-industrial offset values are given, determined by the mean temperature difference between cases at equal $CO_2$ (using a log interpolation between 280 and 560 ppm, to estimate the missing $E^{400}$). A similar offset, corrected for the difference between the $E^{280}$ and $E^{280,P}$, is shown between brackets.

## 4.2 Simulated mid-Pliocene Conditions

### 4.2.1 Atmosphere

Looking at the annual mean near surface air temperature in the $Eoi^{400}$ versus $E^{280}$ case (Figure 4 a,b), we can identify the direct influence of many of the changes made in the PlioMIP2 mid-Pliocene boundary conditions (see also Figure S7 for a side-by-side comparison between the $E^{280}$, $E^{280,P}$, and $Eoi^{280}$ cases). The strongest temperature deviations are seen over parts of Antarctica and Greenland in the absence of land ice. Relatively cool temperatures in the $Eoi^{400}$ can be found over East Antarctica, where we have implemented a thicker ice sheet compared to the pre-industrial conditions. We see similar temperatures between the $E^{280}$ and $Eoi^{400}$ over much of the land at low latitudes, that are mainly a result of surface properties in the Pliocene boundary conditions that counteract the overall warming (e.g. lakes in Africa, vegetation in Australia, see also supplementary Figure 1). Warmest month mean temperatures of $>40°C$ occur over many low and middle latitude continental regions, while coldest month mean temperature contours steadily migrate poleward in the $Eoi^{400}$. As shown in Table 1, we see a considerable polar amplification in the overall warming pattern between the $Eoi^{400}$ and $E^{280}$ with mostly small temperature differences in the tropics increasing to $>10°C$ over both the Arctic and Southern Ocean.

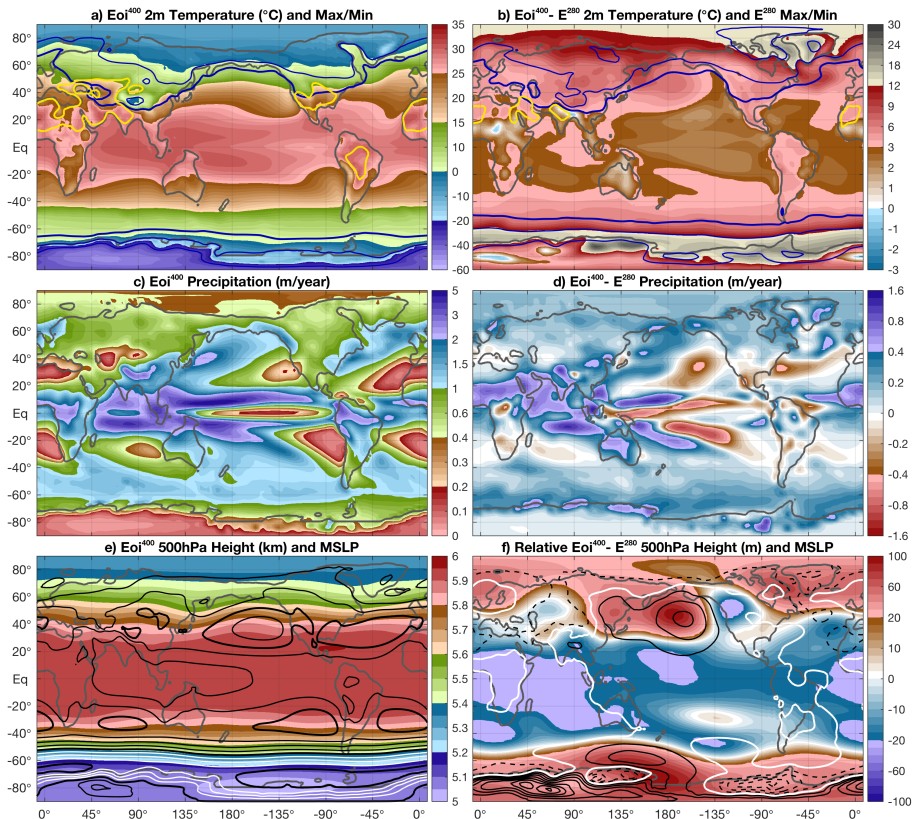

**Figure 4.** Annual mean atmospheric model fields from our Eoi[400] case (left) and the Eoi[400]- E[280] difference. **a,b)** Near surface (2m) air temperature (shading), warmest month mean maximum (yellow contour at 40°C), and coldest month mean minimum (blue contours at 0°C, -20°C, and -40°C). Contours in (a): Eoi[400]; in (b) E[280]. **c,d)** Precipitation (m/year), differences in (d) are smoothed over land. **e)** Height of the 500hPa pressure level (shading) and mean sea level pressure (MSLP; contours every 5hPa, black ≤1000 hPa, white <1000hPa, thick line every 20hPa). **f)** 500hPa height difference with respect to the global average Eoi[400]- E[280] value (i.e. 96m), contours are used for MSLP (every 2 hPa, solid >0, dashed < 0, thick white line at 0hPa).

Besides being warmer, the Eoi[400] is substantially wetter compared to the E[280] (Figure 4 c,d) with a global average annual precipitation of 1180mm versus 1042mm (i.e. a 13%; see Table 1). Polar regions are overall wetter, but most noticeably over the Southern Ocean. Reduced ice sheet cover over much of Antarctica and warmer surface temperatures both act to
increase precipitation over much of the coastal region. An overall poleward migration of storm tracks is seen, which is most pronounced over the North Atlantic Ocean. At lower latitudes, changes in the precipitation pattern are dominated by a shift towards the Eastern Hemisphere. Regions surrounding the Indian and West Pacific Ocean are much wetter in the Eoi[400] while those around the East Pacific and Atlantic Ocean become drier. This is related to a combined westward shift and expansion of the Walker circulation in our mid-Pliocene simulations (shown in most PlioMIP2 simulations by Han et al. 2021). Like many
other comparable model studies, our E[280] shows a double inter-tropical convergence zone (ITCZ) over the Pacific Ocean (e.g.

Bellucci et al. 2010; Haywood et al. 2020). The southern branch of the ITCZ is weaker over the Pacific Ocean in the Eoi[400], while shifting westward and southward towards a pronounced South-Pacific Convergence Zone (SPCZ). A more active Indian Monsoon can explain increased rainfall over the Middle East and South Asia, as well as some of the cooler temperatures in northern India. Higher precipitation rates in the Eoi[400] are also seen over Australia and much of northern Africa, which are

directly related to the altered land surface properties (see also Figure S1 in the supplementary material).

In addition to the direct effects of the PlioMIP2 boundary conditions, some more general changes in the atmospheric circulation are seen between the E[280] and Eoi[400] cases (Figure 4 e,f - note that the global average change in 500hPa height is subtracted in f). A reduction in the equator-to-pole gradient is not only present in surface temperatures, but also in mid-tropospheric height surfaces. This translates to an overall reduction of baroclinic instability, despite some of the increases in precipitation seen at

middle and high latitudes. Reduced surface elevation over much of West Antarctica and Greenland leads to a lower surface pressure in the Eoi[400] while the opposite is seen over parts of East Antarctica. These changes in the surface topography also likely influence the circulation as several stationary ridges are evident in the 500hPa height difference. Preferred midlatitude ridging is evident over the North Pacific Ocean in the Eoi[400], which is in turn reflected in the temperature and precipitation differences with the E[280]. In addition to the advective adjustment (i.e. warmer and wetter west of the high pressure/geopotential

anomaly), the radiative feedback and subsidence act to also increase temperatures at towards the centre of the anomaly.

### 4.2.2 Ocean

Annual mean sea surface temperatures (SSTs) reflect the patterns and changes seen in the Eoi[400] and E[280] atmospheric temperatures (Figure 5 a,b; a similar comparison using the E[280,P] instead can be found in supplementary Figure 8). In contrast to near surface air temperature, the Arctic Ocean SSTs show the smallest difference between the Eoi[400] and E[280] cases despite

large reductions in sea ice cover. Another area with relatively cool temperatures in the Pliocene is the eastern Tropical Pacific Ocean, being still $\sim2^\circ$C warmer compared to the E[280]. Much warmer SSTs are seen over much of the North Atlantic Ocean, as well as the Northwest Pacific Ocean. Apart from the West Antarctic coastal region, the entire Southern Ocean is $\sim4$–8 $^\circ$C warmer in our Pliocene control.

Sea surface salinity (SSS) patterns in the Eoi[400] are mostly driven by precipitation patterns and gateway changes (Figure, 5c,d)

between the E[280]. High precipitation amounts over most of Asia and the Indian Ocean cause low SSS through surface fluxes as well as river runoff. The Arctic Ocean is less saline compared to the E[280] partly as a result of increased runoff, but also due to the closed Bering Strait and Northwest Passage. The outflow of these low salinity Arctic surface waters can be seen to the east and south of Greenland, resulting in a rather complex interaction with much more saline waters in the northern Atlantic Ocean. The Pacific Ocean is in general more saline in the Eoi[400], with regional changes from the E[280] that are the combined effect of

precipitation patterns, enhanced river runoff, and the Bering Strait closure.

The depth averaged flow, represented by the barotropic stream function (BSF) in Figure 5 e,f is very similar between the Eoi[400] and E[280] cases. Regardless of the reduced meridional gradient in surface temperatures, the Antarctic Circumpolar Current (ACC) is slightly stronger in our Pliocene control. In contrast to what is seen at the surface, the depth-averaged density gradient is steeper in the Eoi[400] and therefore enhances the density-driven component of the ACC. Southern Hemisphere subtropical

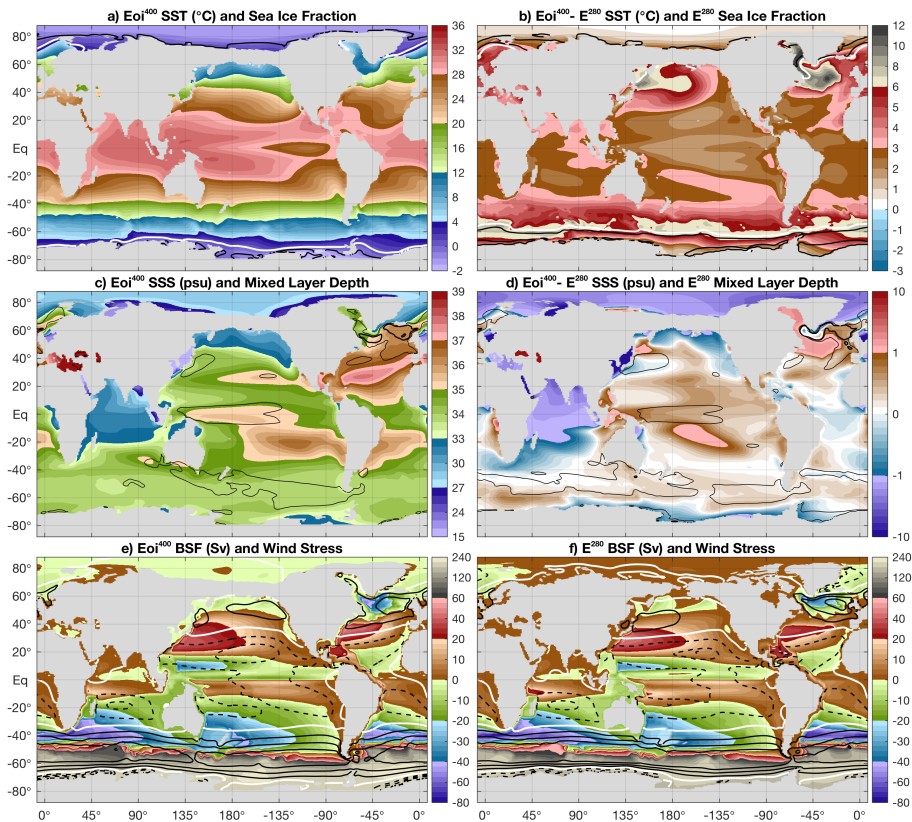

**Figure 5.** Annual mean oceanic model fields from our Eoi$^{400}$ case (left) and the Eoi$^{400}$- E$^{280}$ difference. **a,b)** sea surface temperature temperature (shading) and sea ice fraction (contours; white: 0.15, black: 0.5 and 0.9). Contours in (a): Eoi$^{400}$; in (b) E$^{280}$. **c,d)** Sea surface salinity (shading) and mixed layer depth (thin line: 100m, thick black: 250m, thick white: 500m). **e,f)** Barotropic stream function (BSF; shading) and zonal wind stress (contours every $5 \cdot 10^{-2}$Pa, dashed: $<0$, thick white line at 0hPa). Contours in (e): Eoi$^{400}$; in (f) E$^{280}$.

gyres are slightly weaker, while those in the Northern Hemisphere show a poleward extension in the Eoi$^{400}$. North Atlantic subtropical and subpolar gyres are both stronger compared to the E$^{280}$.

### 4.2.3 Sea ice

Even with the relatively simple model set-up used here, our E$^{280}$ simulation has realistic sea ice cover compared to late 20$^{th}$ century observations from Peng et al. (2013) as shown in Figure 6. The mean sea ice maximum extent corresponds well with the observed modern day maximum in both hemispheres, but the minimum is overestimated especially in the Southern Hemisphere. The disagreement may, however, be mostly explained by the difference between pre-industrial and modern conditions.

Sea ice cover is drastically reduced in our Eoi$^{400}$ simulation compared to the E$^{280}$ one. During the late summer minimum extent, sea ice concentrations drop to nearly zero across both polar regions, with only some remaining over exposed waters in West Antarctica. Changes in the maximum sea ice cover are less dramatic over the Arctic region, while the Southern Ocean

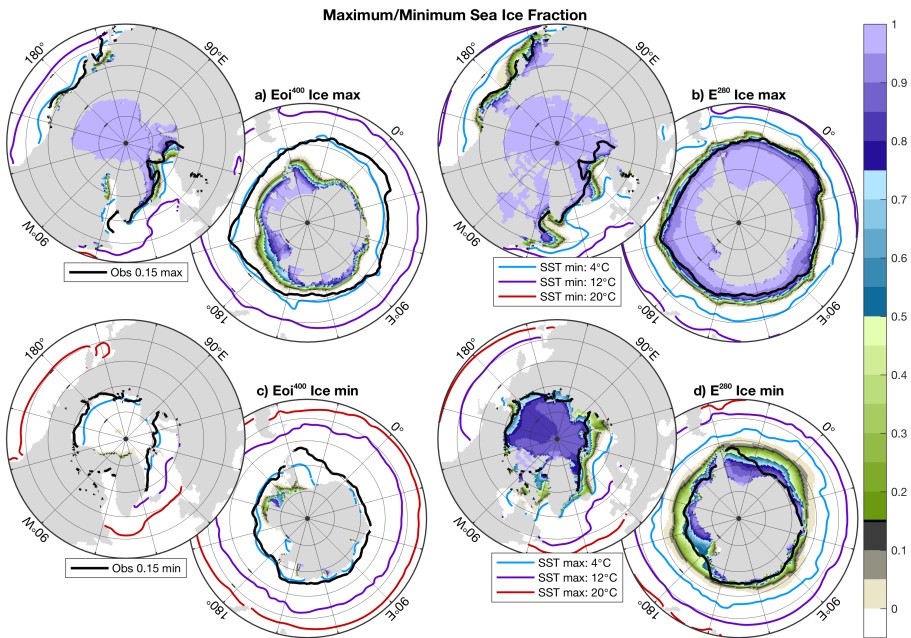

**Figure 6. a)** Eoi[400] mean maximum monthly sea ice fraction over northern and southern polar regions. Coloured contours show coldest month mean SST, the black contour indicates the late 21[st] century observed September (NH)/March (SH) sea ice edge at an 0.15 fraction (indicated on the colour bar; Fetterer et al. 2017). **b)** As in (a), but for the E[280] case using the same contour intervals and sea ice edge data. **c-d)** Similar to (a-b) showing the monthly sea ice minimum, warmest month mean SST and late 21[st] century observed March (NH)/September (SH) sea ice edge.

still has much less sea ice cover in the Eoi[400] compared to the E[280] case. This corresponds well with the ∼4–8°C warmer SSTs seen across the Southern Ocean in our Pliocene simulations (Figure 5b). The isolated nature of the Arctic Ocean allows its surface waters to remain relatively cool, with some sea ice persisting until late summer. Despite substantial refreezing during wintertime, open waters in the early winter months prevent the air from cooling down further. This explains the much larger near surface air temperature anomalies over the Arctic region between our Eoi[400] and E[280] simulations (Figure 4b).

## 4.3 Meridional overturning and heat transport

### 4.3.1 Global

Our Eoi[400] case is characterised by a stronger and deeper global meridional overturning circulation (MOC) compared to the E[280]. This is reflected in the global meridional overturning stream function (MSF) in Figure 7a,b by the dominant northern overturning cell north of 40°S. The deep southern (i.e. negative) overturning cell linked to Antarctic bottom waters is slightly stronger in the Eoi[400] as well. Note that the appearance of this overturning cell is mostly masked in the Southern Ocean by strong Ekman upwelling linked to the ACC. The Eoi[560] MOC is similar to the Eoi[400] one, but overall slightly weaker, while

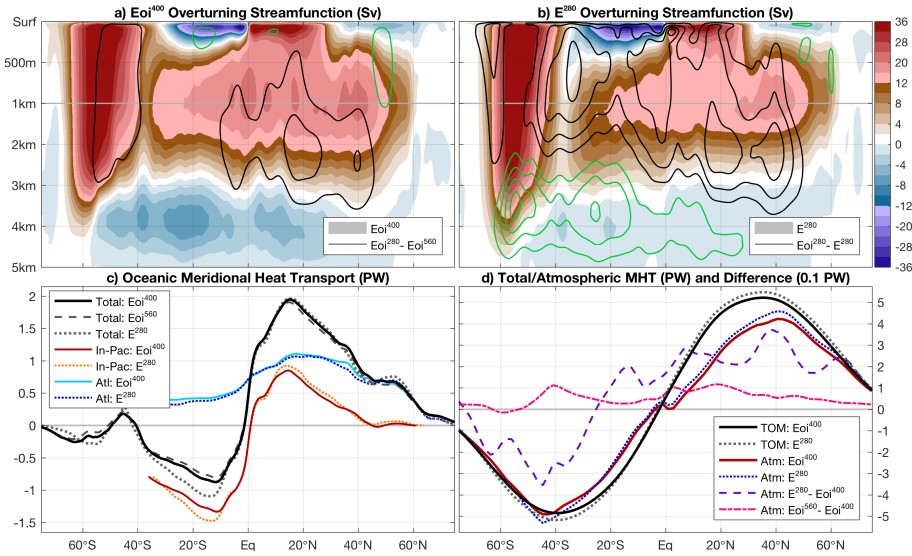

**Figure 7. a)** Global oceanic meridional overturning stream function (MSF) of the Eoi$^{400}$ case (shading) and Eoi$^{280}$- Eoi$^{560}$ difference (contours every 2Sv; black: positive and green: negative), taking the mean over the last 500 model years. **b)** As in (a), but for the E$^{280}$ case and Eoi$^{280}$- E$^{560}$ difference. Note the vertical stretching of the upper 1km. **c)** Oceanic meridional heat transport in the Eoi$^{400}$ (solid lines; global: black, Indo-Pacific: red, and Atlantic Ocean: blue), E$^{280}$ (dashed lines, similar colouring), and Eoi$^{560}$ (dotted line, only global). **d)** Total meridional heat transport, induced by the top of model (TOM) net radiative fluxes (Eoi$^{400}$: solid black, E$^{280}$: dotted grey) and the corresponding atmospheric heat transport (Eoi$^{400}$: solid red, E$^{280}$: dotted blue). The difference in atmospheric heat transport with respect to the Eoi$^{400}$ is shown for the E$^{280}$ (dashed pink) and Eoi$^{560}$ (dash-dotted pink), magnified tenfold.

the opposite holds for the Eoi$^{280}$. This can be partly a transient effect of the simulations, but suggests a tendency of the MOC to get weaker towards higher atmospheric pCO$_2$.

The differences in MSF pattern and strength between the Eoi$^{400}$ and E$^{280}$ cases are only partly reflected in the corresponding
oceanic heat transports (OHT), shown in Figure 7c. Apart from a slight weakening of poleward heat transport at southern low latitudes (mostly in the Indo-Pacific sector), the Eoi$^{400}$ OHT is very similar to the E$^{280}$. The warmer Eoi$^{560}$ case has an overall reduced poleward oceanic heat transport, but the relative change is small (<0.1PW). The total meridional heat transport (MHT, shown in Figure 7d), is slightly weaker in the Eoi$^{400}$ compared to the E$^{280}$. This is a direct response the reduced meridional temperature gradient seen in the Eoi$^{400}$, demanding a lower poleward heat transport on both hemispheres. As there are only
minor changes in the oceanic component, the reduced total MHT must be mainly accounted for by the atmosphere. This is reflected by the atmospheric MHT difference of E$^{280}$ with respect to the Eoi$^{400}$, having mostly the same sign as the total MHT. The only exception is seen at southern low latitudes, where a stronger southward OHT is compensated by a northward atmospheric MHT in the E$^{280}$. The Eoi$^{560}$ atmospheric MHT shows a small net positive offset with respect to the Eoi$^{400}$, that may be explained by some remnant warming trend. In agreement with a further reduction of the meridional temperature

gradient, the warmer Eoi$^{560}$ case thus has a further reduced poleward MHT albeit small ($\sim$0.1 W m$^{-2}$) and mostly accounted for by the ocean.

### 4.3.2    Atlantic Ocean

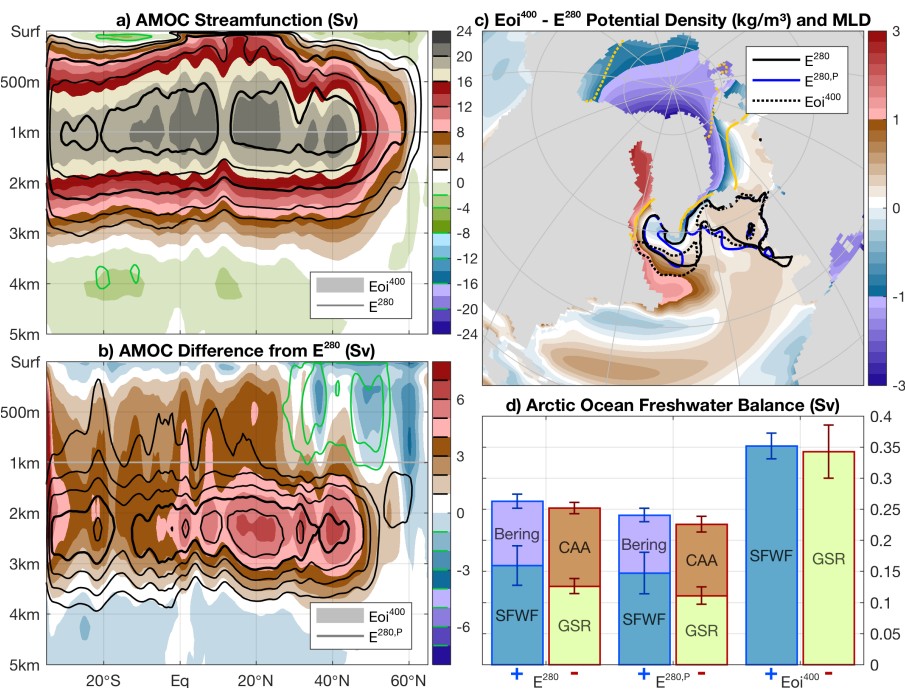

**Figure 8. a)** Mean AMOC stream function over the last 500 years of the Eoi$^{400}$ (shading) and E$^{280}$ simulation (contours highlighted on the colourbar). **b)** Difference in AMOC stream function with respect to the E$^{280}$, for the Eoi$^{400}$ (shading) and the E$^{280,P}$ (contours). **c)** Eoi$^{400}$- E$^{280}$ (100-year mean) potential density difference, averaged over the upper 100m and relative to the global average change. Contours show the annual mean maximum mixed layer depth for the E$^{280}$ (solid black), E$^{280,P}$ (solid blue), and Eoi$^{400}$ (dotted black) and 0.15 sea ice fraction (yellow; solid: E$^{280}$, dashed: Eoi$^{400}$). **d)** Freshwater budget over the Arctic Ocean for the (left to right) E$^{280}$, E$^{280,P}$, and Eoi$^{400}$ cases, including the net surface freshwater flux (SFWF) and transports across the Bering Straight, Canadian Archipelago (CAA), and Greenland-Scotland Ridge (GSR). The mean over the last 500 model years is taken, with the standard deviation of yearly averages indicated by the error bars.

    The Atlantic Meridional Overturning Circulation (AMOC; Figure 8a) clearly encompasses the entire deep northern overturning cell in the global MSF north of 30°S (Figure 7a). The AMOC mean state is overall stronger in our Pliocene versus pre-industrial simulations, but its temporal variability is also considerably higher in all of the Pliocene cases (22.7 $\pm$ 1.6 Sv

versus 18.3 $\pm$ 0.7 Sv; see also Figure S5 in the supplementary material). Especially the Eoi$^{560}$ AMOC undergoes a series of large ($>$10Sv) intensity swings, although this is likely caused by the thermal adjustment of the deep ocean and therefore mostly a transient model feature. These large swings stopped after $\sim$1000 model years, but the AMOC variability remains the largest of any of our cases ($\sigma = 2.4$Sv). The E$^{280,P}$ AMOC also exhibits a larger variability ($\sigma = 1.0$Sv) compared to the

other pre-industrial cases, but still smaller than any of the Pliocene ones. A more detailed discussion on AMOC variability is provided in Section 4.6.2

The difference in AMOC streamfunction between $Eoi^{400}$ and $E^{280}$ is most pronounced between 1 and 3 km depth (Figure 8b). The stronger and deeper AMOC cell can be partly explained by the altered vertical mixing parameters in our Pliocene model set-up, as the $E^{280,P}$ shows a reduced but very similar difference pattern with respect to the $Eoi^{400}$. This is consistent with the

stronger AMOC in the *POP1 type* vertical mixing simulations of Chandan and Richard Peltier (2017). In general, we find that the altered vertical mixing scheme in our Pliocene simulations does have an impact on the strength and behaviour of the AMOC, but not to the extent that is seen in any of the Pliocene cases. It is therefore likely that the altered boundary conditions and resulting circulation changes have a considerable impact on the AMOC strength and behaviour. A stronger mid-Pliocene AMOC was found consistently in the PlioMIP2 ensemble by Zhang et al. (2021), who also found a link with higher North At-

lantic SSTs, but no clear relation with Atlantic OHT. The latter is investigated in more detail by Weifenbach et al. (in prep), who decompose the OHT into the contributions from the overturning and gyre circulations, respectively. The PlioMIP2 boundary conditions applied here include the closure of several high latitude oceanic gateways. This results in lighter upper ocean waters across the Arctic Ocean in the $Eoi^{400}$, but denser waters across most of the North Atlantic Ocean (Figure 8c). The integrated net surface freshwater flux over the Arctic Ocean more than doubles in the $Eoi^{400}$ case compared to the $E^{280}$ one (Figure 8d).

This more than compensates for the missing transport from the Bering straight closure. Lacking connectivity through the Canadian Archipelago demands for an equally large net southward freshwater transport across the Greenland-Scotland ridge. The outflow of those light, low salinity Arctic waters pushes the deep water formation zone towards the south across the Labrador Sea into warmer and deeper waters. Increased salinity in the Labrador Sea and thus higher potential densities are tied to the closure of the Canadian Archipelago, as shown by the negative component in the $E^{280}$ Arctic Ocean freshwater balance (i.e.

net salt transport from Labrador Sea). Such a southward shift of the deep water formation zone is not present in the $E^{280,P}$ case.

## 4.4    Model-proxy comparison

As shown by Haywood et al. (2020), our $Eoi^{400}$ case performs well when comparing the annual mean SSTs to the available PlioMIP2 time-specific proxy records (Foley and Dowsett, 2019; McClymont et al., 2020). The zonally averaged, annual mean SST from our different simulations is shown in light of these proxies in Figure 9a. Considering site-specific rather than zonally

averaged SSTs from the $Eoi^{400}$ simulations, we find a root mean square error (RMSE) of 2.7°C and a mean absolute error (MAE) of 2.2°C (using the combined $U_{37}^{K'}$ and Mg/Ca proxy record of McClymont et al. 2020). Moreover, there is no significant warm or cold model bias across the SST/latitude range covered by the proxies. Relatively warm North Atlantic SSTs in the model are well reflected by the proxy record, consistent with the other CCSM/CESM models within PlioMIP2 as shown by de Nooijer et al. (2020). They also find that the CCSM/CESM model family best captures mid-Pliocene Arctic SST proxies,

which is likely related to low sea ice cover.

Looking at the SST difference between Pliocene and pre-industrial conditions, rather than absolute values, reveals some more discrepancies between proxy records and our $Eoi^{400}$ simulation (Figure 9b). Most of the high positive temperature anomalies at northern middle and high latitudes remain well captured by the model, as well as the much smaller differences at low latitudes.

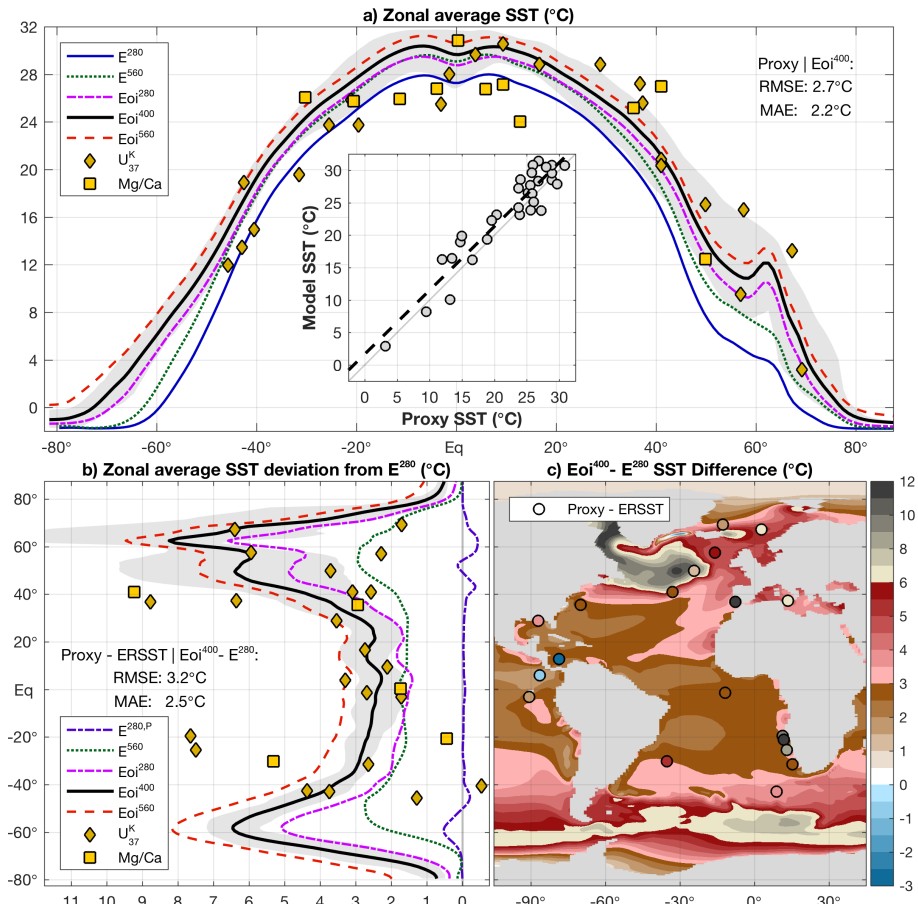

**Figure 9. a)** Proxy-based annual mean SST reconstructions (yellow markers; diamonds: $U_{37}^{K'}$, squares: Mg/Ca) versus model-derived zonally averaged SST for $Eoi^{400}$ (black, including zonal variation in grey), $E^{280}$ (blue), $E^{560}$ (green dotted), $Eoi^{280}$ (pink dash-dotted), and $Eoi^{560}$ (orange dashed). The inset shows a point-wise comparison, including a simple linear regression. **b)** Pliocene proxy - pre-industrial SST (ERSST; late 19$^{th}$ century reanalysis) and modelled zonal average SST difference from $E^{280}$ (colouring as in a; purple dash-dotted: $E^{280,P}$). **c)** Pliocene - pre-industrial SST difference over the Atlantic Ocean: proxy - pre-industrial (coloured markers) and model-derived $Eoi^{400}$- $E^{280}$ (shading).

However, other sites at both northern and southern middle latitudes show a much poorer agreement between model-based and proxy indicated Pliocene - pre-industrial SST differences. The lack of proxy data across most of the Southern Ocean makes it impossible to assess whether the large SST differences between our Pliocene and pre-industrial simulations are realistic. Despite the visually poor agreement compared to absolute mid-Pliocene SSTs, the RMSE and MAE are only slightly higher for the Pliocene - pre-industrial SST difference.

Surprisingly large Pliocene - pre-industrial SST differences (considering a limited $CO_2$ effect) were found in the PlioMIP1, especially over the North Atlantic Ocean, but not well reflected in the model ensemble (Haywood et al., 2013b). This seems to be

greatly improved now, as shown by the large positive anomalies over much of the middle to high latitude North Atlantic Ocean in Figure 9c. The largest discrepancies between proxy-based and model-derived Pliocene - pre-industrial SST differences are seen in the Mediterranean Sea and off the coasts of northern America and southern Africa. These discrepancies can be largely explained by coastal upwelling and boundary currents, which are poorly resolved in our simulations because of the limited

horizontal resolution (e.g. McClymont et al. 2020; Li et al. 2019). Although absolute Pliocene SSTs seem to agree better, they are already much too warm at some of these locations in our $E^{280}$ case compared to reanalysis data.

Near surface air temperatures reveal the same warming pattern between the $Eoi^{400}$ and $E^{280}$ case as seen in the SSTs over low and middle latitudes (see also Figure S9 in the supplementary material). The highest temperature anomalies shift poleward, over the Arctic Ocean and the Antarctic coastal region due to sea ice melt and reduced ice sheet cover. Our coolest

Pliocene simulation ($Eoi^{280}$) is similar or even warmer than the $E^{560}$ case at all latitudes. This indicates the importance of the implemented mid-Pliocene model boundary conditions and long-term impacts such as the partial loss of ice sheets.

### 4.5   Sensitivity to Pliocene boundary conditions versus atmospheric $pCO_2$

Using our set of pre-industrial and Pliocene *sensitivity* simulations (i.e. $E^{280}$, $E^{560}$, $Eoi^{280}$, and $Eoi^{560}$), we can make an assessment of the effects of the applied Pliocene boundary conditions ($Eoi^n$- $E^n$) versus external radiative forcing ($X^{560}$- $X^{280}$), as well

as any state-dependency in the model response. The difference in annual mean near surface air temperature is shown in Figure 10. The top panels of this figure show the modelled temperature response to a shift from pre-industrial to Pliocene boundary conditions (excluding $CO_2$), the bottom ones to a doubling atmospheric $pCO_2$. Consequently, comparing left to right panels shows the state-dependency of the effect of boundary conditions to a different $pCO_2$ baseline (top), and vice versa (bottom).

Some generic temperature differences between the model cases can be easily identified in all of these comparisons. Stronger

contrasts are seen over land than over the ocean, as a result of the thermal capacity and much larger potential for latent heat fluxes over the ocean. Polar amplification of the temperature change also seems universal across our simulations, caused mainly by ice-albedo feedbacks and reversed lapse rate feedbacks (i.e. positive at high latitudes, negative at low latitudes; see also Figure S7 in the supplementary material). As noted before, the ECS of our simulated Pliocene climate is very similar to that of the pre-industrial one (i.e. 3.2°C versus 3.17°C per $CO_2$ doubling). It is therefore not surprising that the overall response to

a doubling of atmospheric $pCO_2$ is comparable, with mainly some regional differences as a result of the boundary conditions and local feedback mechanisms.

Changes in precipitation patterns between the four *sensitivity* simulations show large differences between the response to atmospheric $pCO_2$ and other model boundary conditions. In line with the findings of Han et al. (2021), the effect of a $CO_2$ doubling is mainly a wet gets wetter/dry gets drier-response and focusses in the tropics. Differences in precipitation between Pliocene and

pre-industrial cases are generally much more pronounced and widespread, but are also enhanced between the high versus low $CO_2$ scenarios. Some of these differences are directly related to the boundary conditions (mainly topography and ice sheets) and their temperature response (e.g. sea ice, monsoons). In addition, large-scale precipitation changes include a westward and poleward shift in the tropics resulting in a wetter tropical Indian and West Pacific Ocean in our mid-Pliocene simulations. Han et al. (2021) show that these dynamic changes in precipitation are related to the meridional and zonal circulation patterns

(e.g. Walker Circulation influences the Indian-Pacific moisture exchange). Moreover, our *CCSM4-Utr* simulations are found to exhibit the largest asymmetry in hemispheric energy flux within the PlioMIP2 ensemble, explaining the significant shifts of the Pacific ITCZ and SPCZ. Further north, we see a poleward migration of North Atlantic storm tracks and the dynamic response of a prevailing North Pacific anticyclone. These shifts in precipitation correspond well with what is seen in Figure 4d.

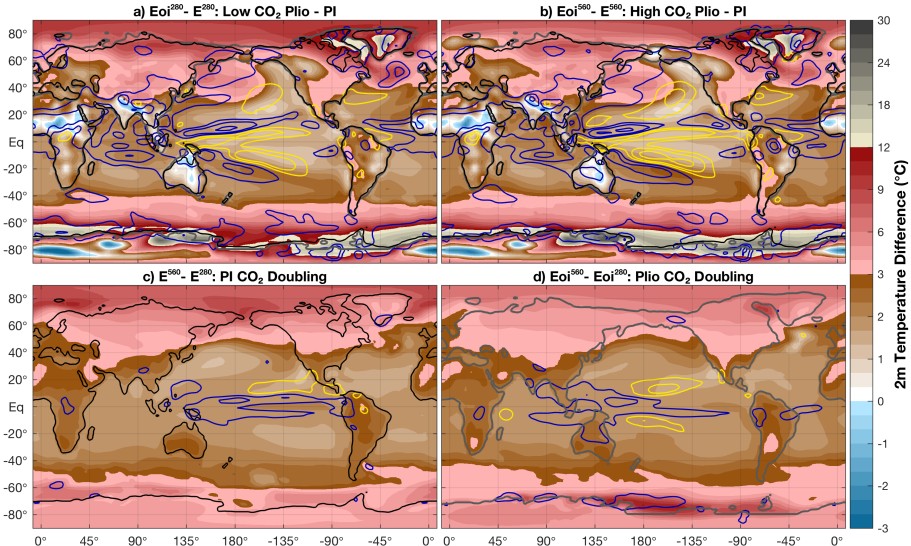

**Figure 10.** Annual mean near surface air temperature difference between our four *sensitivity* simulations; **a)** Eoi$^{280}$- E$^{280}$, **b)** Eoi$^{560}$- E$^{560}$, **c)** E$^{560}$- E$^{280}$, and **d)** Eoi$^{560}$- Eoi$^{280}$. Contours showing the difference in mean annual precipitation at 100, 200 and 500 mm; blue: positive and yellow: negative.

We also consider the zonally averaged temperature differences between the four *sensitivity* cases, along with the corresponding energy balance model (EBM) analysis and the contribution from its different components (Figure 11). All of the zonally averaged temperature responses estimated using the EBM are in near perfect agreement with the actual surface temperature differences in the model. Simply adding the contributions from albedo, emissivity and MHT reproduces the full temperature response well, showing that nonlinear effects between them are small. Both these findings are in line with the results of Hill

et al. (2014) for PlioMIP1. Albedo plays a crucial role in the response to mid-Pliocene boundary conditions, especially at the surface (i.e. vegetation, ice sheets, and sea ice). The effect of planetary albedo is about a third lower compared to the surface, showing that the latter is partly compensated by the shortwave contribution from clouds, aerosols and water vapour. The emissivity of longwave radiation shows a more similar response between the effect of boundary conditions vs. CO$_2$ doubling, but is still stronger in the former. About half of the temperature contribution through emissivity after a CO$_2$ doubling is related to the

direct radiative forcing, leaving the other half as mainly the result of water vapour and lapse rate feedbacks. These feedbacks, together with the surface albedo, are thus the main drivers behind polar amplification of the temperature difference between all of the different cases considered here. The effect is partly mitigated by meridional heat transport and cloud feedbacks, but

these have little impact on the global scale. This is again consistent with the findings of Hill et al. (2014), with a larger effect of the mid-Pliocene boundary conditions in PlioMIP2 versus PlioMIP1.

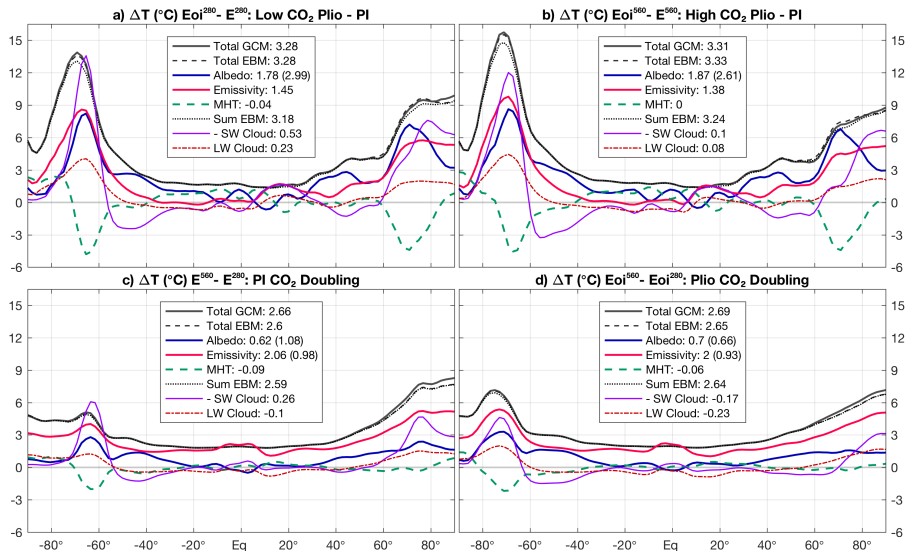

**Figure 11.** As in Figure 10, but showing zonally averaged temperature differences and contributions from different components within the energy balance model (EBM). The temperature difference between our simulations (Total GCM; solid gray) is compared to the one predicted by the EBM (Total EBM; dashed dark grey). Components of the EBM include: planetary albedo (solid blue), emissivity (solid red), meridional heat transport (MHT; dashed green), shortwave cloud forcing (negative; solid purple) and longwave (dashed dark red) cloud forcing. The sum of the contributions from albedo, emissivity and MHT is also shown (Sum EBM; dashed black). Globally averaged values are shown for each component, with bracketed values added for surface albedo and emissivity excluding the effect of $CO_2$ doubling.


Most of the effects of the Pliocene model boundary conditions on temperature (Figure 10 a,b) agree well with those seen earlier between our $E^{280}$ and $Eoi^{400}$ cases, shown in Figure 4b. The warming effect of removing/lowering ice sheets at high latitudes and cooling effect of introducing lakes or removing desert at low latitudes are independent of the atmospheric $CO_2$ level. Over the Arctic Ocean, reduced sea ice in the Pliocene simulations has an increased impact on temperatures above the

surface towards lower $pCO_2$ (Figure 10 a vs. b). The opposite is seen over parts of the Southern Ocean, where sea ice cover is still relatively large in the $E^{560}$ (see also Figure S10 in the supplementary material). Temperature changes that can be related to altered circulation patterns in the Pliocene versus pre-industrial (e.g. monsoons, South Pacific Convergence Zone, midlatitude ridging, storm tracks) are robust between different $CO_2$ levels. The temperature response to a doubling of atmospheric $pCO_2$ is very similar at low and middle latitudes between our pre-industrial and Pliocene simulations (Figure 10 c,d). Differences in sea

ice cover (and to a lesser extent land ice and surface properties) amplify high latitude warming in response to a $CO_2$ doubling, most prominently over the Arctic in the pre-industrial cases and over the Southern Ocean in the Pliocene ones.

The near surface air temperature differences between our four *sensitivity* cases (Figure 10) are reflected in the SST (see Figure

S10 in the supplementary material). The most prominent exception to this consistency is found over high latitude surface waters, which are seen to change much less in temperature than the air above. This can be explained by the decoupling effect of sea

ice, which tends to dampen the SST differences (but with large seasonal differences). Pliocene - pre-industrial SST differences over the Pacific Ocean are consistent between $CO_2$ levels and noticeably different from the effect of a $pCO_2$ doubling. These patterns suggest a shift in the background state of both ENSO and PDO in our Pliocene simulations (see also Section 4.6). An even more prominent and consistent SST response to the Pliocene boundary conditions is found over the North Atlantic Ocean (also seen in Figure 5b). This is likely the combined result of a stronger Pliocene AMOC and the isolation of the Labrador Sea

through closure of the Canadian Archipelago. Large changes in SST between our Pliocene and pre-industrial simulations can be linked to the upwelling of relatively warm deep waters south of the ACC, and reflected in the air temperatures above (Figure 10).

## 4.6 Modes of internal variability

### 4.6.1 El Niño Southern Oscillation

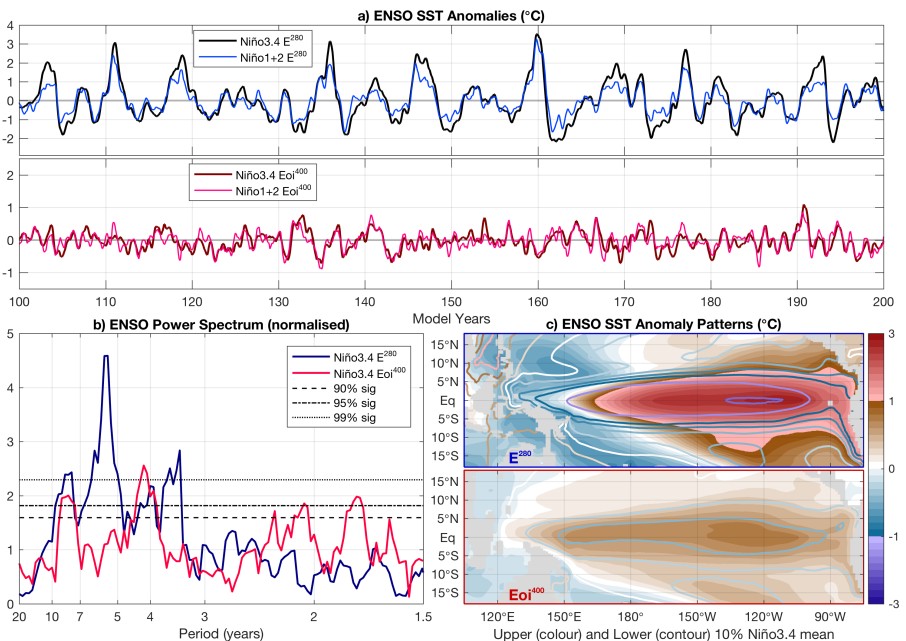

**Figure 12. a)** ENSO time series for the $E^{280}$ (blue) and $Eoi^{400}$ (red), using monthly SST anomaly fields and a 5-month running mean. Note the different scaling; temperature intervals are kept the same for visual comparison. **b)** Multi-taper power spectrum, using 200 years of monthly data including 90%, 95% and 99% confidence levels based. **c)** Corresponding ENSO SST patterns, taking the mean over the 10% highest (shading) and 10% lowest (contours) monthly Niño 3.4 index values.

Not only the mean state, but several modes of variability in our simulated Pliocene climate are also different from what is seen in the pre-industrial reference. As it plays a primary role in the tropics, we first look at the behaviour of the El Niño Southern Oscillation (ENSO). In Figure 12, time series of the Niño 1+2 and Niño 3.4 indices are shown for the $E^{280}$ and $Eoi^{400}$ case. The amplitude of both ENSO indices is greatly reduced in our $Eoi^{400}$ versus $E^{280}$ case ($\sigma_{1+2}$: -54%; $\sigma_{3.4}$: -68%). This is the largest such reduction seen among the PlioMIP2 ensemble (Figure 2a of Oldeman et al. 2021; OB21). Furthermore, the

occurrence of strong and long-lasting ($>1°C$; $>1$ year) El Niño/La Niña events completely disappears in our mid-Pliocene simulation. Both these findings are consistent among all of our mid-Pliocene (i.e. $Eoi^{280}$, $Eoi^{400}$, $Eoi^{560}$) versus pre-industrial (i.e. $E^{280}$, $E^{280,P}$, $E^{560}$, $E^{1120}$) cases (not shown).

Spectral analysis of the Niño 3.4 indices also shows large differences between the ENSO behaviour in the $E^{280}$ and $Eoi^{400}$ case. The modelled pre-industrial ENSO variability is characterised by a broad spectral peak at periods of 3–10 years, with 3

statistically significant (at 99% confidence) peaks and a dominant period around 6 years. A similar broad peak is seen for the mid-Pliocene ENSO, but only with significant variability at 4 years (99%) and 9 years (95%). Moreover, there is an increase of significant variability in the $Eoi^{400}$ case at shorter periods around 2 years or less. This agrees well with the predominantly weak and rather high frequent behaviour of the modelled Pliocene ENSO, seen in the Niño 3.4 time series. A similar drop in spectral power of the mid-Pliocene ENSO is seen consistently in the PlioMIP2 ensemble, but more focussed at the 3–5 year

period (Figures 3 and 4 of OB21).

As to be expected from both of the Niño time series shown for each case, the amplitude of SST anomalies is seen to be much smaller throughout the tropical Pacific Ocean in our $Eoi^{400}$ versus $E^{280}$ case. Looking at the mean SST pattern corresponding to the 10% average highest and lowest monthly Niño 3.4 values allows us to assess both the spatial distribution and strength of ENSO variability at the same time. In addition to an overall highly reduced amplitude, the associated region of SST variability

expands westward and poleward across the Central/West Pacific Ocean in our Pliocene simulation. This is consistent with the pattern shifts shown by the majority PlioMIP2 models (Figure 5 of OB21). The mean difference in SST between the $Eoi^{400}$ and $E^{280}$ (Figure 5b) over the main region of ENSO variability is smaller than the average tropics. The background state of the tropical Pacific Ocean is therefore not El Niño-, but rather slightly La Niña-like. Only few models within the PlioMIP2 ensemble show such a strong La Niña-like pattern in the mean temperature of the $Eoi^{400}$, but these are also the models with the

largest reduction in ENSO variability (see also Figures 9-11 of OB21).

### 4.6.2   Interannual to multi-decadal SST variability

Using 500 years of annual mean SSTs and AMOC strength, we assess the dominant modes of variability in the North Pacific and North Atlantic Ocean (Figure 13). We consider the dominant eigenmode of SST anomalies in the North Pacific Ocean to represent the PMV, and the one best correlating with average North Atlantic SSTs as the AMV. The explained variance by the

PMV mode is seen to decrease in the $Eoi^{400}$ versus $E^{280}$ case, while that of the AMV greatly increases. As noted earlier, the AMOC variability is considerably larger in the $Eoi^{400}$ case compared to the $E^{280}$ one. Correlations between the PMV, AMV, AMOC and annual Niño 3.4 indices can be found in Figure S12 in the supplementary material.

Spectral analysis shows that the $E^{280}$ PMV has a broad peak at 10–20 years and therefore strongly resembles the Pacific Decadal

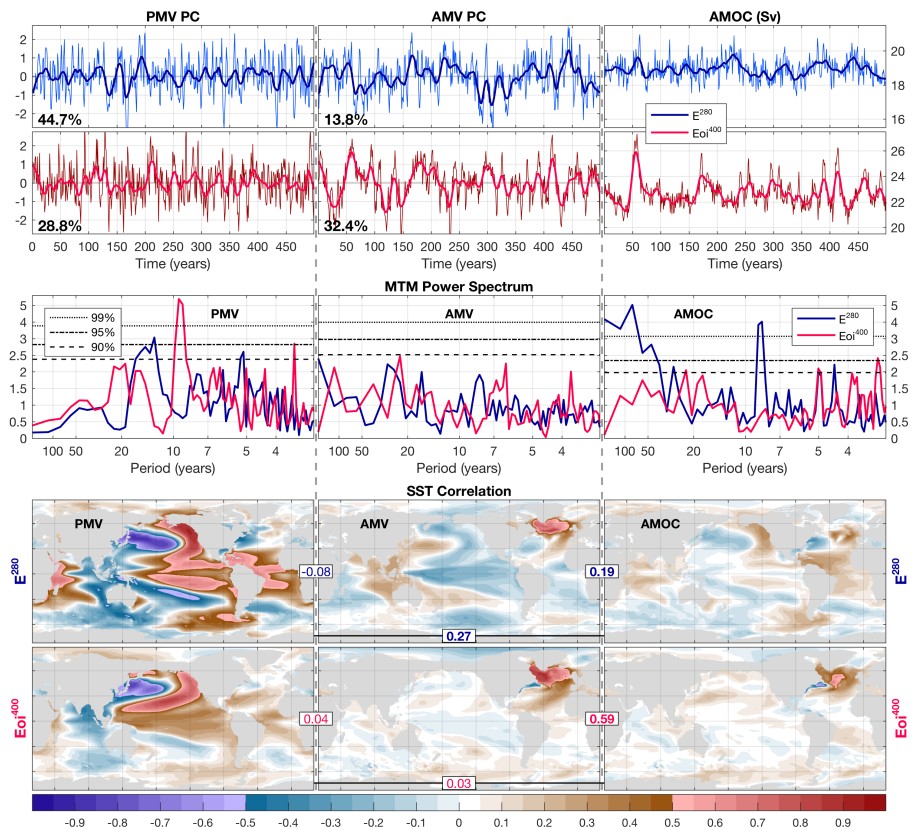

**Figure 13.** Time series (upper panels; thick lines using a 20-year smoothing window), multi-tapered power spectra (middle panels), and spatial correlation patterns (lower panels) of North Pacific SST (left), North Atlantic SST (middle), and AMOC strength (right) variability, for 500 years of annual SST data from our $E^{280}$ (blue) and $Eoi^{400}$ (red) simulation. The percentage of variance explained by the respective principal component is shown in black along with the PMV and AMV time series. Spatial SST patterns show the temporal correlation between the SST at each location and the different variability indices. Boxed numbers also indicate the total correlation between each of these indices (i.e. time series in the upper panels.)

Oscillation (PDO). This broad peak is seen to shift towards longer periods for the $Eoi^{400}$ PMV, which is now accompanied by a sharper peak at just under 10 years. Looking at the patterns of SST variability associated to the PMV, an almost global footprint in the $E^{280}$ is reduced to mostly a North Pacific signature in the $Eoi^{400}$. This is likely related to the strongly decreased ENSO variability in the $Eoi^{400}$ versus $E^{280}$ case, as both the low latitude SST patterns and the spectral properties at periods <10 years show a high resemblance between PMV and ENSO in both simulations. This is supported by significant correlations ($E^{280}$: 0.48; $Eoi^{400}$: 0.34) between the PMV and yearly Niño 3.4 indices. The role of the AMV on global SST variability is much more limited compared to PMV, but again strongest and more widespread in the $E^{280}$ case. Meanwhile, the $Eoi^{400}$ has a much higher percentage of explained variance and is strongly connected to AMOC variability. The spatial pattern of SST anomalies associated to the AMOC strength shows little resemblance to either the PMV or AMV in the $E^{280}$ case, but both their time

series are significantly (yet weakly) correlated. In contrast, the AMOC strength strongly correlates with the AMV for the Eoi[400] and their associated spatial patterns show a large resemblance. It is thus clear that the stronger and more variable AMOC in our

Pliocene simulations has a profound impact on not only the mean North Atlantic SSTs, but also their variability. Additionally, it seems that the different modes of SST variability are less connected between ocean basins in the Eoi[400] versus E[280] case.

## 5   Conclusions

We have completed a set of simulations using the CCSM4/CESM1.0.5 within the PlioMIP2, including E[280], E[560], Eoi[280], Eoi[400], and Eoi[560] cases (E[280,P] and E[1120] as additional sensitivity experiments). Our simulations show a warm mPWP climate,

which is 4.7°C warmer compared to our pre-industrial control in terms of globally averaged near surface air temperature. With an estimated ECS of 3.1°C per $CO_2$ doubling, which is consistent throughout the different pre-industrial and Pliocene cases, much of the mPWP signal is thus related to the effect of the applied PRISM4 boundary conditions. The effect of those boundary conditions is mainly to increase temperatures by about 3°C on average globally and 2.4°C in the ocean. The vertical distribution of oceanic temperatures is altered in our Pliocene simulations by the use of enhanced background vertical diffu-

sivity. Our E[280,P] simulation can explain most of this differential warming pattern between pre-industrial and Pliocene cases. The enhanced vertical mixing acts to warm the deep ocean at the expense of the upper ocean, without affecting the globally averaged surface temperature. Most of the other results are thus not significantly influenced by the altered vertical diffusivity, with only regional differences seen at the surface and that become negligible throughout the rest of the atmosphere.

The combined effect of increased atmospheric $pCO_2$ and altered boundary conditions not only make our Eoi[400] case warmer than the E[280] one, but also induce substantial polar amplification. This is mainly the result of latitudinally-dependent surface albedo (ice and vegetation), water vapour and lapse rate feedbacks. Cloud feedbacks are small in the global average, with the shortwave component acting to mitigate high latitude warmth. The temperature differences are most pronounced where ice sheets are removed (including a large elevation effect) and high latitude oceans, where sea ice cover is greatly reduced in

all of the Pliocene cases. Especially the shortwave radiative feedbacks resulting from the mid-Pliocene boundary conditions are much stronger compared to those from a $CO_2$ doubling. Furthermore, the latitudinal dependence in all of the considered radiative components is stronger, helping to explain the relatively warm high latitude regions in our Pliocene simulations. The Eoi[400] climate shows an increased precipitation rate with respect to the E[280] one. Both equatorial and high latitude regions are generally wetter in the Pliocene, while the sub-tropics become drier (with a distinct poleward shift of the storm tracks).

Particularly high precipitation amounts are seen over most of the Indian Ocean and it surroundings, owing to enhanced North African and South/Southeast Asian monsoons in the Pliocene.

The warmer Eoi[400] climate with a reduced meridional temperature gradient compared to the pre-industrial reference agrees well with the available proxies. A strong Pliocene warming signal over the North Atlantic Ocean is seen in both the proxy record and our simulations, which can be linked to an enhanced AMOC. This stronger AMOC, however, is not linked to an overall

increase in oceanic meridional heat transport. Moreover, the total (atmosphere + ocean) top of model induced meridional heat

transport is slightly reduced in the warmer simulations, in line with the reduced meridional temperature gradient. The stronger and more variable AMOC in our Pliocene versus pre-industrial simulations can in part be explained by the altered vertical diffusivity parameter. Yet, it is mostly the result of the applied mid-Pliocene boundary conditions, in particular the closure of several Arctic gateways.


In addition to differences in the mean state between our $Eoi^{400}$ and $E^{280}$ cases, there are clear shifts in the different modes of variability studied here: ENSO, PMV and AMV. The ENSO amplitude is greatly reduced in our Pliocene simulations and characterised by shorter periodicity compared to the pre-industrial reference. The corresponding spatial pattern is also spread out across much of the tropical Pacific Ocean. Closely related to ENSO is the PMV in our $E^{280}$ case, both having a distinct

fingerprint on global SST variability on various time scales ranging from annual to multi-decadal. This teleconnection is lost in the $Eoi^{400}$ case, with the PMV influence being mostly confined to just the North Pacific Ocean. Meanwhile, the AMV shows a strong connection to AMOC variability in our Pliocene simulations. Their mutual influence seems to be the dominant source of SST variability in the $Eoi^{400}$, as opposed to ENSO/PMV in the $E^{280}$.

As most of the PRISM4 boundary conditions, which are applied here as an external forcing to the model simulation, are in fact

the result of long term feedbacks (i.e. ice melt and vegetation changes), the $Eoi^{400}$ can serve as a good analog for future climatic changes. Our simulations show not only a strong warming compared to the pre-industrial reference, but also considerable regional changes and shifts in the dominant modes of variability.

*Data availability.* PlioMIP2 model data, including those of the simulations presented here, can be downloaded from the server located at the School of Earth and Environment of the University of Leeds. Contact Alan Haywood (a.m.haywood@leeds.ac.uk) for access. The

last 100 model years of our $E^{280}$, $E^{560}$, $Eoi^{280}$, $Eoi^{400}$, and $Eoi^{560}$ are available within the dataset. PlioMIP2 data from CESM2, EC-Earth3.3, NorESM1-F, IPSLCM6A and GISS2.1G can be obtained through the Earth System Grid Federation (ESGF) (https://esgf-node.llnl.gov/search/cmip6/, last access: 14 October 2021, ESGF, 2021).

*Author contributions.* MAK, MLJB, and ASvdH set up the model simulations, MAK and MLJB managed the simulations and post-processing of the data. MLJB performed the analyses and set up the manuscript. All authors contributed to the final shape and contents of the manuscript.

*Competing interests.* The authors declare to have no competing interests.

*Acknowledgements.* This work was carried out under under the program of the Netherlands Earth System Science Centre (NESSC), financially supported by the Ministry of Education, Culture and Science (OCW, grant #024.002.001). Simulations were performed at the

SURFsara dutch national computing facilities and were sponsored by NWO-EW (Netherlands Organisation for Scientific Research, Exact Sciences) under the projects 17189 and 2020.022.

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
