# Peer review of "Warm mid-Pliocene conditions without high climate sensitivity: the CCSM4-Utrecht (CESM 1.0.5) contribution to the PlioMIP2"

_Climate of the Past, 2021_

## Referee Comment (RC1)

**Summary**

This paper presents the contribution to PlioMIP from the CCSM4-Utrecht (CESM1.0.5) model. The broad-scale features of the Pliocene simulation are presented, and in addition there is a model-data comparison, a factorisation analysis of the CO2 versus non-CO2 boundary conditions, and the modes of variability are explored. Overall, I think that this is a nicely written and presented paper, and will likely be of benefit to other group in PlioMIP who will find it useful when interpreting other results from the wider PlioMIP ensemble. However, it is somewhat descriptive, and at times it is a little speculative as to the mechanism involved, but this is the nature of a paper such as this, so I think this is OK.

**Main comments**

(M1) In the abstract and in Section 3.2, it is proposed that the relative warmth of the Pliocene simulation compared with other PlioMIP models is the initialisation and long spinup. This may be true, but it would be good if this could be verified more robustly, for example by explicitly presenting and comparing the integration lengths and initial conditions of all models in PlioMIP, and/or showing the Utrecht global mean temp after a similar amount of spinup as other models, for a direct comparison.

(M2) In Section 4.6 it would be good to have more of a direct comparison with the results of Oldemann et al (in press), - try to build on their results in this section.

(M3) Similarly in the section on ocean circulation (4.3) I would expect to see here an in-depth comparison with Zhang et al (2021), and here to bring additional insights, and to note how this model fits in with the larger ensemble.

(M4) Line 91-99 – if the vertical diffusivity makes little or no difference to the model results, as is claimed, then why did you modify them in the Pliocene? This needs to be better explained and justified. I would expect to maps of the temperature difference between these two different model versions, at least in Supp info.

(M5) Section 4.5 - Here, I think the paper would benefit from use/discussion of the factorisation framework presented in Lunt et al (2021), for analysing these simulations. For example, the mean of Figure 10 (top left and top right) could be presented.

(M6) Section 4.4 – I would recommend using the McClymont et al SSTs instead of Foley and Dowsett, because McClymont et al have been peer-reviewed.

(M7) Line 263 – 272 – careful here. I am not sure that I agree with this interpretation of the changes in fluxes. If both simulations are in equilibrium, then both simulations will have a net zero energy balance at the surface and TOA. Interpreting a change in shortwave net flux is not necessarily an indicator of changes in feedbacks. A full energy balance analysis (e.g. Heinemann et al, 2009; Hill et al, 2014) or even better, a APRP analysis would be more appropriate here.

(M8) section 4.3.2 - Rather than just presenting SST and surface temperature (which are very similar), why not show the same analysis but for e.g. precipitation, or seaice, which may be more interesting?

**Specific Comments**

(S1) Figure 1 – for the modern ice sheet, it seems odd to me that there are large parts of Antarctica that are not ice covered (see light blue contour) but are above sea level (see colour scale). I would have expected the whole Antarctic continent to be covered in an ice sheet (which it is, according to figure S1).

(S2) Figure 2 – what happens at ~1000 years? The model appears to be taking in energy before this time, and then releases heat. Any idea why?

(S3) Line 180 – It is not just slow feedbacks that can give a non-linearity, it is simply the intrinsic non-linear nature of all feedbacks, especially clouds; see e.g. Bloch-Johnson et al., (2015) or Knutti et al. (2015).

(S4) Line 229-231 – "The globally averaged sea surface temperature (SST) only increases by 2.1 $^oC$ per $CO_2$ doubling, as a result of the inhomogeneous distribution of land/sea surface" – This is perhaps more to do with lack of snow-cover and icesheet (and seaice to a certain extent) feedbacks for the SSTs, and lack of evaporation over land; i.e. it is a result of the well-known land-sea contrast in warming.

(S5) Line 235 – 241 – This section could benefit from some literature around the non-linearity of forcings/feedbacks. Could also give a feedback parameter (units W/m2 K-1)

(S6) I am not sure that the discussion of surface versus deep ocean temperature is robust given the different mixing coefficients in the simulations (see comment M1).

(S7) Line 287 – "This is in agreement with a larger ice volume over parts of East Antarctica" . I am not sure I follow the mechanism here – why is this in agreement?

(S8) Line 305 – there does seem to be a coincidence with maximum warming and mslp/500mbar geopotential height, but the reason for this coupling is not clear- one might expect a longitudinal shift in the temperature response so that it coincided with the anomalous north/south winds, rather than the centre of the geopotential anomaly?

(S9) Section 4.2.3, Figure 6. For the seaice observations, if the model were perfect then which fraction of seaice would lie on the observed contour line? 100%, 0%, or 50%?

**Technical Comments**

(T1) Figure 4,5 – show absolute of both E280 and Eoi400, and the difference – there is room for 3 plots side-by-side if the full page-width is used.

(T2) Figure 7 – be consistent throughout whether Eoi400 is on the left or right (left here, right in figure 6)

(T3) Line 24 – relatively stable

(T4) Line 29 – foe -> for

(T5) Line 37 – cite Haywood et al (2020) large scale features of PlioMIP2.

(T6) Line 52 – is it really equivalent to the latest version? This implies you are using the latest CMIP6 version, which is not the case I believe.

(T7) Line 65 – "switching to an adjusted Pliocene climatology"

(T8) Line 165 – "Within the PlioMIP2" – database?

(T9) Line 285 – besides *being* warmer

Review by: Dan Lunt

**References**

Zhang, Z., Li, X., Guo, C., Otterå, O. H., Nisancioglu, K. H., Tan, N., Contoux, C., Ramstein, G., Feng, R., Otto-Bliesner, B. L., Brady, E., Chandan, D., Peltier, W. R., Baatsen, M. L. J., von der Heydt, A. S., Weiffenbach, J. E., Stepanek, C., Lohmann, G., Zhang, Q., Li, Q., Chandler, M. A., Sohl, L. E., Haywood, A. M., Hunter, S. J., Tindall, J. C., Williams, C., Lunt, D. J., Chan, W.-L., and Abe-Ouchi, A.: Mid-Pliocene Atlantic Meridional Overturning Circulation simulated in PlioMIP2, Clim. Past, 17, 529543, https://doi.org/10.5194/cp-17-529-2021, 2021.

Oldeman, A. M., Baatsen, M. L. J., von der Heydt, A. S., Dijkstra, H. A., Tindall, J. C., Abe-Ouchi, A., Booth, A. R., Brady, E. C., Chan, W.-L., Chandan, D., Chandler, M. A., Contoux, C., Feng, R., Guo, C., Haywood, A. M., Hunter, S. J., Kamae, Y., Li, Q., Li, X., Lohmann, G., Lunt, D. J., Nisancioglu, K. H., Otto-Bliesner, B. L., Peltier, W. R., Pontes, G. M., Ramstein, G., Sohl, L. E., Stepanek, C., Tan, N., Zhang, Q., Zhang, Z., Wainer, I., and Williams, C. J. R.: Reduced El Nino variability in the mid-Pliocene according to the PlioMIP2 ensemble, in press, Clim. Past.

Lunt, D. J., Chandan, D., Haywood, A. M., Lunt, G. M., Rougier, J. C., Salzmann, U., Schmidt, G. A., and Valdes, P. J.: Multi-variate factorisation of numerical simulations, Geosci. Model Dev., 14, 43074317, https://doi.org/10.5194/gmd-14-4307-2021, 2021.

Bloch-Johnson, J., Pierrehumbert, R. T., & Abbot, D. S. (2015). Feedback temperature dependence determines the risk of high warming. Geophysical Research Letters, 42, 4973–4980. Retrieved from https://doi.org/10.1002/2015GL064240 .

Knutti, Reto & Rugenstein, Maria. (2015). Feedbacks, climate sensitivity and the limits of linear models. Philosophical Transactions of the Royal Society A: Mathematical, Physical and Engineering Sciences. 373. 20150146. 10.1098/rsta.2015.0146.

Hill, D. J., Haywood, A. M., Lunt, D. J., Hunter, S. J., Bragg, F. J., Contoux, C., Stepanek, C., Sohl, L., Rosenbloom, N. A., Chan, W.-L., Kamae, Y., Zhang, Z., Abe-Ouchi, A., Chandler, M. A., Jost, A., Lohmann, G., Otto-Bliesner, B. L., Ramstein, G., and Ueda, H.: Evaluating the dominant components of warming in Pliocene climate simulations, Clim. Past, 10, 79-90, [doi:10.5194/cp-10-79-2014]

Heinemann, M., Jungclaus, J. H., and Marotzke, J.: Warm Paleocene/ Eocene climate as simulated in ECHAM5/MPI-OM, Clim. Past, 5, 785–802, doi:10.5194/cp-5-785-2009, 2009.

Haywood, A. M., Tindall, J. C., Dowsett, H. J., Dolan, A. M., Foley, K. M., Hunter, S. J., Hill, D. J., Chan, W.-L., Abe-Ouchi, A., Stepanek, C., Lohmann, G., Chandan, D., Peltier, W. R., Tan, N., Contoux, C., Ramstein, G., Li, X., Zhang, Z., Guo, C., Nisancioglu, K. H., Zhang, Q., Li, Q., Kamae, Y., Chandler, M. A., Sohl, L. E., Otto-Bliesner, B. L., Feng, R., Brady, E. C., von der Heydt, A. S., Baatsen, M. L. J., and Lunt, D. J.: The Pliocene Model Intercomparison Project Phase 2: large-scale climate features and climate sensitivity, Clim. Past, 16, 2095-2123, https://doi.org/10.5194/cp-16-2095-2020, 2020.

---

## Referee Comment (RC2)

**General Comments**

This manuscript presents new simulations of the Pliocene warm period using the CESM model. The authors present simulations using a range of different CO2 levels, using both modern and Pliocene boundary conditions. They find significant warming due to changing the boundary conditions, mainly because of ice-albedo effects that allow a larger insolation, independently of greenhouse forcing. The model's climate sensitivity to CO2 is roughly the same under both boundary conditions. The model achieves a generally very good fit to Pliocene proxies, and the remaining discrepancies are examined in an appropriate manner. The paper is mostly about describing the model and its main features of variability, and it is generally well written, so I suggest mainly minor revisions to clarify the data presentation.

My biggest recommendation for change is to revise the colormaps in the anomaly plots. I suspect the authors have put significant effort into the color schemes, so I'm sorry to insist upon changes here. However, I find that the color scheme used for most of the manuscript figures does (a) a good job of representing absolute values, and (b) a poor job of representing anomalies. There are several reasons for this:

The banded color regions tend to create "critical values" when changing to different colours. This is ok when there is no particular critical threshold in the data, but with anomaly plots, there is a critical value of zero that must be highlighted. Having 6 different colour bands in the anomaly scale means there seem to be critical values jumping out everywhere, and it's hard to get an intuitive sense of the positive and negative changes. The second reason is that some colours have a highly suggestive nature that can be deceptive. For example, most papers use red for a warm anomaly and blue for a cold anomaly, which makes intuitive sense. The authors have in many places used blue shading for warm anomalies, which is very jarring to interpret. (E.g. Fig 4b, 5b, 9c, 10, 11). I suggest for all of the anomaly plots (especially temperature and precipitation) either use:

A) only one colour (with intensity shading) either side of the zero value, so that the critical values are very obvious, e.g. red for warming, blue for cooling;
B) use two colours either side of the zero, but choose them to be carefully matching in tone and intuitive, e.g. purple and blue for cooling, brown and red for warming. Or: green and blue for wetting, brown and red for drying.

Apart from this, I have a couple of scientific suggestions:

1. Why is there a large change in direction of the temperature trends at around 1000 years in the Eoi400 run? This is a curious feature of the spinup that deserves a stronger explanation.

2. Since the main result is that Pliocene boundary conditions cause significant warming (independently of CO2), it would be good to examine the radiative forcing changes in more detail. This can be done using a framework such as in Lunt et al (2021, https://doi.org/10.5194/cp-17-203-2021) and
Heinemann et al (2009, https://doi.org/10.5194/cp-5-785-2009)

**Line Comments**

L29: "foe" typo

L93: This equation looks a bit ugly in current format. Is it possible to use nicer labels, such as "d" for depth rather than "dpth", and why do "vdc1" and "vdc2" need so many characters? Why not "c1" and "c2" for instance, and use subscripts for a nicer appearance?

L182: TOM has not been defined in the main text. It was defined in a Figure caption but it should be spelled out in the main text as well.

L210-211: "to not select a mode?" is a strange way of phrasing this. Are the authors trying to say that they (a) calculated EOFs for the North Atlantic, and then (b) disregarded leading EOF modes that correlated highly with ENSO or the PMV? I don't understand, please clarify.

L219: "more easy" → easier

L321: "Straight" → Strait

Figure 5 caption: I think it's better to use "variables" rather than "observables"

Figure 7a,b: There is too much information stacked in the overturning plots. The contours can't be seen properly on top of the colours. I suggest expanding this plot to put the Eoi560 overturning on separate panels - there is plenty of space to do so.

L357: "clearly reflected atmospheric MHT difference": there's a word missing here, please clarify

Figure 8a,b: Again please expand the overturning plots to use separate plots for different streamfunctions. The contours are too difficult to read over the colours - it is information overload.

Figure 9c: Here the use of blue to signify warming is really jarring, especially the blue proxy circles. Please revise the anomaly colorbars (as in my general comments).

L409-410: Here it might be useful to reference Li et al (2019, https://doi.org/10.1029/2019PA003760) which shows the impact of changes to coastal upwelling on large-scale Pliocene SSTs

L414: This sentence would be improved by deleting "It is noteworthy that"

Figure 10: As noted above on colorbars: there are large swathes of blue used to represent warm anomalies. Please revise.

Figure 12c: The contours overlaid on colours here are very difficult to interpret (as in Figs 7, 8). Please expand the number of panels to separate the clashing information.

L483: "there is a lot more": perhaps delete "a lot", since this a vague descriptor.

L523-524: "this differential warming patterns" : fix grammar. Also, instead of saying "different parameter choice", can you be more specific and say "enhanced diffusivity"?

L532: "dryer" → drier

---

## Author Comment (AC2)

**Figures supporting author comments on CP 2021-140**

[Figure]

*Figure C1: Energy balance analysis following Lunt et al. (2021), zonal average differences.*

[Figure]

*Figure C2: Altered version of Figure 10, with simplified colour scale and precipitation contours.*

[Figure]

*Figure C3: Updated version of Figure 7, using contours for differences only.*

---

## Author Response (AR1)

**Response to reviewer 1**

Summary

This paper presents the contribution to PlioMIP from the CCSM4-Utrecht (CESM1.0.5) model. The broad-scale features of the Pliocene simulation are presented, and in addition there is a model-data comparison, a factorisation analysis of the CO2 versus non-CO2 boundary conditions, and the modes of variability are explored. Overall, I think that this is a nicely written and presented paper, and will likely be of benefit to other group in PlioMIP who will find it useful when interpreting other results from the wider PlioMIP ensemble. However, it is somewhat descriptive, and at times it is a little speculative as to the mechanism involved, but this is the nature of a paper such as this, so I think this is OK.

*AC: The authors would like to thank Dr. Dan Lunt for the detailed feedback and comments. We mostly agree with the main remark that some of the analyses were too qualitative and have made a number of improvements there. We also want to point out that the main goal of this manuscript is to look at the results of our model results, albeit within the PlioMIP2 ensemble. We therefore have chosen to not present any new analyses of model data beyond our specific set of simulations, but rather refer to other relevant PlioMIP2 studies as much as possible. A number of recent PlioMIP2 studies were added in the references and discussion.*

Main comments

(M1) In the abstract and in Section 3.2, it is proposed that the relative warmth of the Pliocene simulation compared with other PlioMIP models is the initialisation and long spinup. This may be true, but it would be good if this could be verified more robustly, for example by explicitly presenting and comparing the integration lengths and initial conditions of all models in PlioMIP, and/or showing the Utrecht global mean temp after a similar amount of spinup as other models, for a direct comparison.

*AC: The main point made here is that the model was initialised with an average ocean temperature from a PlioMIP1 CCSM4 simulation, which has a very similar model set-up. Our Eoi400 simulation still warms up considerably, indicating the importance of an extended (>1kyr) model spin-up. This is why we also show the complete time series of our model spin-up phase. We do not see much added value comparing the spin-up procedures of all of the PlioMIP simulations within the scope of this manuscript. This is now better motivated and explained in sections 2.4 and 3.2, including a reference to Chandan et al. (2017.*

(M2) In Section 4.6 it would be good to have more of a direct comparison with the results of Oldemann et al (in press), - try to build on their results in this section.

*AC: We have improved the connection to Oldeman et al. (2021) here (Figure 2a: standard deviation, Figure 4b: spectral shift, Figure 5: pattern shift), as they show that our simulation has the largest reduction in ENSO amplitude between E280 and Eoi400 cases within the PlioMIP2 ensemble.*

(M3) Similarly in the section on ocean circulation (4.3) I would expect to see here an in-depth comparison with Zhang et al (2021), and here to bring additional insights, and to note how this model fits in with the larger ensemble.

*AC: We now include a more extensive (albeit mostly) qualitative comparison to the results of Zhang et al (2021) here, as the deepening and/or strengthening of the AMOC in the Pliocene seems to be robust within the PlioMIP ensemble. A more in-depth analysis of the underlying mechanisms and contribution of the AMOC to meridional heat transports will be presented by Weiffenbach et al. (in prep.), which is now mentioned as well.*

(M4) Line 91-99 – if the vertical diffusivity makes little or no difference to the model results, as is claimed, then why did you modify them in the Pliocene? This needs to be better explained and justified. I would expect to maps of the temperature difference between these two different model versions, at least in Supp info.

*AC: A direct comparison between a pre-industrial simulation with/without mixing adjustment is made in sup. Figures 4, 7 and 9. The first of these shows how the vertical distribution of heat in the ocean is altered, but surface temperatures are left mostly unchanged (globally averaged). Figure S7 adds a side-by-side comparison of the pre-industrial reference, the effect of our altered vertical diffusivity, and the Pliocene boundary conditions at constant CO2. Figure S9 repeats the pre-industrial to mid-Pliocene comparison of Figure 5, but using the reference with mixing adjustment instead. The choices made regarding the vertical diffusivity are now better motivated and explained. The original Eoi400 simulation, which was uploaded to the PlioMIP2 database had the modified vertical mixing parameters. We therefore keep using this simulation as the standard and have added the sensitivity simulations with other vertical mixing configurations to the supplement.*

(M5) Section 4.5 - Here, I think the paper would benefit from use/discussion of the factorisation framework presented in Lunt et al (2021), for analysing these simulations. For example, the mean of Figure 10 (top left and top right) could be presented.

*AC: Our set of simulations is not elaborate enough to carry out the suggested sensitivity analyses. However, as suggested by reviewer 2, we implemented an energy balance model analyses similar to Heinemann et al. (2009), also used by Hall et al. (2014) for PlioMIP1 and Lunt et al (2021) for DeepMIP simulations. We have replaced the current table and discussion of direct fluxes by the results of this EBM decomposition, which is explained in section 3.3. The results of the EBM analysis are shown in Figure 11, which will replace Table S2, and discussed in section 4.5 which has been largely re-written.*

(M6) Section 4.4 – I would recommend using the McClymont et al SSTs instead of Foley and Dowsett, because McClymont et al have been peer-reviewed.

*AC: We now make use of the McClymont et al. (2020) SST proxies, using a combination of their UK37- and Mg/Ca-based estimates. Figure 9 was and the discussion in section 4.4 have both been updated.*

(M7) Line 263 – 272 – careful here. I am not sure that I agree with this interpretation of the changes in fluxes. If both simulations are in equilibrium, then both simulations will have a net zero energy balance at the surface and TOA. Interpreting a change in shortwave net flux is not necessarily an indicator of changes in feedbacks. A full energy balance analysis (e.g. Heinemann et al, 2009; Hill et al, 2014) or even better, a APRP analysis would be more appropriate here.

*AC: This part has been replaced by the EBM analysis figure and discussion in section 4.5.*

(M8) section 4.3.2 - Rather than just presenting SST and surface temperature (which are very similar), why not show the same analysis but for e.g. precipitation, or seaice, which may be more interesting?

*AC: Figure 8 has been adjusted, now showing also sea ice extent. Precipitation differences between the sensitivity cases have been added to Figure 10. The corresponding figure showing sea surface temperatures is now in the supplement.*

Specific Comments

(S1) Figure 1 – for the modern ice sheet, it seems odd to me that there are large parts of Antarctica that are not ice covered (see light blue contour) but are above sea level (see colour scale). I would have expected the whole Antarctic continent to be covered in an ice sheet (which it is, according to figure S1).

*AC: This was an error in the land mask, which has been corrected to better show the actual extent of the ice sheets.*

(S2) Figure 2 – what happens at ~1000 years? The model appears to be taking in energy before this time, and then releases heat. Any idea why?

*AC: During the first part of the spin-up, there is only a shallow and sluggish AMOC. A much stronger and deeper northern overturning cell only materialises after those first ~1000 years, greatly impacting the heat distribution in the ocean and global heat budget. This is now explained here and a reference to Figure S5 added, which shows the temporal evolution of the AMOC strength.*

(S3) Line 180 – It is not just slow feedbacks that can give a non-linearity, it is simply the intrinsic non- linear nature of all feedbacks, especially clouds; see e.g. Bloch-Johnson et al., (2015) or Knutti et al. (2015).

*AC: This part has been adjusted, adding the suggested references.*

(S4) Line 229-231 – "The globally averaged sea surface temperature (SST) only increases by 2.1 oC per CO2 doubling, as a result of the inhomogeneous distribution of land/sea surface" – This is perhaps more to do with lack of snow-cover and icesheet (and seaice to a certain extent) feedbacks for the SSTs, and lack of evaporation over land; i.e. it is a result of the well-known land-sea contrast in warming.

*AC: This is indeed the first-order effect, which we did not state clearly here. The main point was that, on top of this effect, the land-sea distribution further enhances the contrast between globally averaged temperatures over land versus sea. This is now clarified in the text.*

(S5) Line 235 – 241 – This section could benefit from some literature around the non-linearity of forcings/feedbacks. Could also give a feedback parameter (units W/m2 K-1)

*AC: We now refer to Caballero & Huber 2013, Baatsen et al. 2021, Lunt et al. 2021) and link to the added EBM analysis.*

(S6) I am not sure that the discussion of surface versus deep ocean temperature is robust given the different mixing coefficents in the simulations (see comment M1).

*AC: The effect of the mixing coefficients is taken into account here, looking at the crosses in Figure 3. This actually explains some seemingly inconsistent differences in deep ocean temperature between the pre-industrial and mid-Pliocene cases. This is now better explained, Figure 3 has been adjusted slightly and we refer to Figure S4.*

(S7) Line 287 – "This is in agreement with a larger ice volume over parts of East Antarctica" . I am not sure I follow the mechanism here – why is this in agreement?

*AC: The discussion of temperature and precipitation got entangled here, they are now separated and clarified.*

(S8) Line 305 – there does seem to be a coincidence with maximum warming and mslp/500mbar geopotential height, but the reason for this coupling is not clear- one might expect a longitudinal shift in the temperature response so that it coincided with the anomalous north/south winds, rather than the centre of the geopotential anomaly?

*AC: The patterns of Z500/MSLP and temperature/precipitation both show a zonal shift at middle latitudes. We typically see higher precipitation at the western flank of high pressure anomalies, lower at the eastern flank. The temperature anomalies are shifted more towards the centre of pressure anomalies, mostly due to the effects of vertical motion, precipitation and radiative feedbacks (on top of meridional advection). Some discussion was added to the text.*

(S9) Section 4.2.3, Figure 6. For the seaice observations, if the model were perfect then which fraction of seaice would lie on the observed contour line? 100%, 0%, or 50%?

Technical Comments

*AC: The late 20th century sea ice edge is indicated by a 15% sea ice concentration, which is indicated in the colour bar. This was not entirely clear, so the colour scale and the figure caption were adjusted to improve this.*

(T1) Figure 4,5 – show absolute of both E280 and Eoi400, and the difference – there is room for 3 plots side-by-side if the full page-width is used.

*AC: Rather than adding more panels here (full width pages unfortunately do not use the full page width after typesetting), Figure S7 has been added showing a side-by-side comparison of the same fields for the E280, E280,P and Eoi280 cases.*

(T2) Figure 7 – be consistent throughout whether Eoi400 is on the left or right (left here, right in figure 6)

*AC: This figure has been rearranged for consistency.*

(T3) Line 24 – relatively stable

*AC: This has been adjested*

(T4) Line 29 – foe -> for

*AC: This has been corrected*

(T5) Line 37 – cite Haywood et al (2020) large scale features of PlioMIP2.

*AC: This reference was added here.*

(T6) Line 52 – is it really equivalent to the latest version? This implies you are using the latest CMIP6 version, which is not the case I believe.

*AC: 'latest' is probably not the right word here, as we want to point out that the earliest versions of CESM1 are identical (with the settings used here) in terms of model components to the 'last' version of CCSM, i.e. CCSM4 and therefore can be referred to as either CESM1 or CCSM4. This has been rephrased for clarity.*

(T7) Line 65 – "switching to an adjusted Pliocene climatology"

*AC: This sentence has been split to improve readability.*

(T8) Line 165 – "Within the PlioMIP2" – database?

*AC: This has been changed to 'PlioMIP2 database, model fields …' and noted which of the model data is available within the database (i.e. last 100 years of the E280, E560, Eoi280, Eoi400 and Eoi560 cases). This has also been added to the 'data availability' statement.*

(T9) Line 285 – besides *being* warmer

*AC: This has been added.*

Review by: Dan Lunt

**Response to reviewer 2**

General Comments

This manuscript presents new simulations of the Pliocene warm period using the CESM model. The authors present simulations using a range of different CO2 levels, using both modern and Pliocene boundary conditions. They find significant warming due to changing the boundary conditions, mainly because of ice-albedo effects that allow a larger insolation, inde- pendently of greenhouse forcing. The model's climate sensitivity to CO2 is roughly the same under both boundary conditions. The model achieves a generally very good fit to Pliocene proxies, and the remaining discrepancies are examined in an appropriate manner. The paper is mostly about describing the model and its main features of variability, and it is generally well written, so I suggest mainly minor revisions to clarify the data presentation. My biggest recommendation for change is to revise the colormaps in the anomaly plots. I suspect the authors have put significant effort into the color schemes, so I'm sorry to insist upon changes here. However, I find that the color scheme used for most of the manuscript figures does (a) a good job of representing absolute values, and (b) a poor job of representing anomalies. There are several reasons for this:
The banded color regions tend to create "critical values" when changing to different colours. This is ok when there is no particular critical threshold in the data, but with anomaly plots, there is a critical value of zero that must be highlighted. Having 6 different colour bands in the anomaly scale means there seem to be critical values jumping out everywhere, and it's hard to get an intuitive sense of the positive and negative changes. The second reason is that some colours have a highly suggestive nature that can be deceptive. For example, most papers use red for a warm anomaly and blue for a cold anomaly, which makes intuitive sense. The authors have in many places used blue shading for warm anomalies, which is very jarring to interpret. (E.g. Fig 4b, 5b, 9c, 10, 11). I suggest for all of the anomaly plots (especially temperature and precipitation) either use:
A) only one colour (with intensity shading) either side of the zero value, so that the critical values are very obvious, e.g. red for warming, blue for cooling;
B) use two colours either side of the zero, but choose them to be carefully matching in tone and intuitive, e.g. purple and blue for cooling, brown and red for warming. Or: green and blue for wetting, brown and red for drying.

*AC: The authors would like to thank the reviewer for the detailed feedback and specific comments. A lot of thought has indeed gone into the colour schemes, but we agree that especially the difference plots can be improved. Having asymmetric (about 0) colour bars in many of the difference plots is motivated by largely one-sided temperature changes. We do agree, that it is best practice to not incorporate blue/green colours on the negative side of the scheme. All of the figures have been reconsidered and adjusted to improve on this part. All of the difference plots have been updated by making the diverging colour bars simpler (only using orange/red and blue/purple shades). All other figures have also been updated to improve the readability of colour schemes and contour intervals.*

Apart from this, I have a couple of scientific suggestions:

1. Why is there a large change in direction of the temperature trends at around 1000 years in the Eoi400 run? This is a curious feature of the spinup that deserves a stronger explanation.

*AC: This indeed stands out in the spin-up of our Eoi400 simulation. In the first phase of this spin-up, there is only a shallow and sluggish AMOC. Only after ~1000 years, we see the development of a much deeper and stronger northern overturning cell which then has a significant impact on the global heat distribution and radiative budget. This is now explained in section 3.2 with a reference to Figure S5, showing the full evolution of the AMOC strength.*

2. Since the main result is that Pliocene boundary conditions cause significant warming (independently of CO2), it would be good to examine the radiative forcing changes in more detail. This can be done using a framework such as in Lunt et al (2021, https://doi.org/10.5194/cp-17-203-2021) and Heinemann et al (2009, https://doi.org/10.5194/cp-5-785-2009)

*AC: We agree that using this framework fits well within this study, using the set of model simulations that we present. This analysis has been added, replacing the straight comparisons of radiative fluxes in sup. Table S2 by a zonal average decomposition in Figure 11. The results lead to similar conclusions, but most of section 4.5 has been re-written in line with the new analysis.*

Line Comments

L29: "foe" typo

*AC: This has been corrected.*

L93: This equation looks a bit ugly in current format. Is it possible to use nicer labels, such as "d" for depth rather than "dpth", and why do "vdc1" and "vdc2" need so many characters?
Why not "c1" and "c2" for instance, and use subscripts for a nicer appearance?

*AC: We chose to follow the exact syntax used in the CESM reference manual and related publications. Although we agree that the equation can be simplified/clarified, we prefer to keep it in the current form for consistency.*

L182: TOM has not been defined in the main text. It was defined in a Figure caption but it should be spelled out in the main text as well.

*AC: TOM is now defined at the first notice on line 194.*

L210-211: "to not select a mode?" is a strange way of phrasing this. Are the authors trying to say that they (a) calculated EOFs for the North Atlantic, and then (b) disregarded leading EOF modes that correlated highly with ENSO or the PMV? I don't understand, please clarify.

*AC: This can indeed be clarified; the EOF related to the AMO can be somewhat tricky to find as the North Atlantic SSTs are also influenced by several external factors such as PDO/ENSO/AMOC. Rather than just taking the first/dominant EOF, we therefore select the one that correlates best with the 10-70N average North Atlantic SST such that we are comparing similar modes between the different simulations. This is explained more clearly in the text now.*

L219: "more easy" → easier

*AC: This has been adjusted.*

L321: "Straight" → Strait

*AC: This has been corrected.*

Figure 5 caption: I think it's better to use "variables" rather than "observables"

*AC: Although we prefer the term observables, to distinguish physically meaningful variables from others used internally by the model, we agree that a variable such as the barotropic stream function is not something one could easily observe directly. We have changed this to 'model fields' in the Figure captions of Figures 4, 5, and S7.*

Figure 7a,b: There is too much information stacked in the overturning plots. The contours can't be seen properly on top of the colours. I suggest expanding this plot to put the Eoi560 overturning on separate panels - there is plenty of space to do so.

*AC: This figure has been simplified to improve readability, now only showing the differences in stream function using contours There was too much information here, not all of which was relevant. Colours now show the Eoi400/E280 overturning stream function, contours the effect of a CO2 doubling vs. mid-Pliocene BCs. Figure 8 has also been updated for consistency.*

L357: "clearly reflected atmospheric MHT difference": there's a word missing here, please Clarify

*AC: This has been changed into 'reflected in the atmospheric MHT difference'*

Figure 8a,b: Again please expand the overturning plots to use separate plots for different streamfunctions. The contours are too difficult to read over the colours - it is information overload.

*AC: This issue is partly solved by switching towards a more simple colour scheme. Showing the Eoi560 and Eoi280 meridional overturning stream functions as well mostly served to point out that the differences between Pliocene and Pre-industrial are mostly because of the topographic changes and mixing parametrisation. Similar to Figure 7, we prefer to just show the Eoi400 and E280 stream functions, and only the Eoi280-Eoi560 and Eoi280-E280 differences in contours.*

Figure 9c: Here the use of blue to signify warming is really jarring, especially the blue proxy circles. Please revise the anomaly colorbars (as in my general comments).

*AC: The revised colour schemes should make this more straightforward.*

L409-410: Here it might be useful to reference Li et al (2019, https://doi.org/10.1029/2019PA003760) which shows the impact of changes to coastal upwelling on large-scale Pliocene SSTs

*AC: This reference was added here.*

L414: This sentence would be improved by deleting "It is noteworthy that"

*AC: This part of the sentence has been removed.*

Figure 10: As noted above on colorbars: there are large swathes of blue used to represent warm anomalies. Please revise.

*AC: The colour schemes have been revised accordingly.*

Figure 12c: The contours overlaid on colours here are very difficult to interpret (as in Figs 7, 8). Please expand the number of panels to separate the clashing information.

*AC: This should be improved by the adjusted colour scheme, the contours have been simplified as well.*

L483: "there is a lot more": perhaps delete "a lot", since this a vague descriptor.

*AC: this can indeed be left out, this has been rephrased to make it clear that there is larger variability rather than it being more significant (the latter is also the case, but not the focus here).*

L523-524: "this differential warming patterns" : fix grammar. Also, instead of saying "different parameter choice", can you be more specific and say "enhanced diffusivity"?

*AC: This part has been adjusted and the diffusivity specified.*

L532: "dryer" → drier

*AC: This has been corrected.*